# Thrombospondin expression in myofibers stabilizes muscle membranes

**Davy Vanhoutte[1], Tobias G Schips[1], Jennifer Q Kwong[1], Jennifer Davis[1], Andoria Tjondrokoesoemo[1], Matthew J Brody[1], Michelle A Sargent[1], Onur Kanisicak[1], Hong Yi[2], Quan Q Gao[3], Joseph E Rabinowitz[4], Talila Volk[5], Elizabeth M McNally[3], Jeffery D Molkentin[1,6]***

[1]Cincinnati Children's Hospital Medical Center, Department of Pediatrics, University of Cincinnati, Cincinnati, United States; [2]Robert P. Apkarian Integrated Electron Microscopy Core, Emory University, Atlanta, United States; [3]Center for Genetic Medicine, Northwestern University, Chicago, United States; [4]Temple University School of Medicine, Philadelphia, United States; [5]Department of Molecular Genetics, Weizmann Institute of Science, Rehovot, Israel; [6]Howard Hughes Medical Institute, Cincinnati Children's Hospital Medical Center, Cincinnati, United States

**Abstract** Skeletal muscle is highly sensitive to mutations in genes that participate in membrane stability and cellular attachment, which often leads to muscular dystrophy. Here we show that Thrombospondin-4 (Thbs4) regulates skeletal muscle integrity and its susceptibility to muscular dystrophy through organization of membrane attachment complexes. Loss of the *Thbs4* gene causes spontaneous dystrophic changes with aging and accelerates disease in 2 mouse models of muscular dystrophy, while overexpression of mouse Thbs4 is protective and mitigates dystrophic disease. In the myofiber, Thbs4 selectively enhances vesicular trafficking of dystrophin-glycoprotein and integrin attachment complexes to stabilize the sarcolemma. In agreement, muscle-specific overexpression of *Drosophila* Tsp or mouse Thbs4 rescues a *Drosophila* model of muscular dystrophy with augmented membrane residence of βPS integrin. This functional conservation emphasizes the fundamental importance of Thbs' as regulators of cellular attachment and membrane stability and identifies Thbs4 as a potential therapeutic target for muscular dystrophy.

*For correspondence: jeff. molkentin@cchmc.org

**Competing interests:** The authors declare that no competing interests exist.

## Introduction

Muscle degenerative diseases such as muscular dystrophy (MD) are most commonly caused by mutations in genes that are part of the dystrophin-glycoprotein (DGC) complex or the integrin complex of proteins (*Grounds et al., 2005*; *McNally and Pytel, 2007*). In addition, proper post-translational processing and trafficking of these complexes to the sarcolemma are essential to form a molecular attachment network between the myofilament proteins within the myofibers and the basal lamina and extracellular matrix (ECM) outside the cell (*Goddeeris et al., 2013*; *Liu et al., 2012*; *Xu et al., 2009*). This attachment network provides critical structural support to the plasma membrane (sarcolemma) to withstand contractile forces (*Burr and Molkentin, 2015*; *Grounds et al., 2005*; *Gumerson and Michele, 2011*; *Lapidos et al., 2004*). When this attachment network is deficient in MD, membrane ruptures occur leading to intracellular calcium influx that causes myofiber necrosis, an inflammatory response, fibrosis and fatty tissue replacement, and ultimately muscle functional loss and death (*Burr and Molkentin, 2015*; *Gumerson and Michele, 2011*; *Lapidos et al., 2004*). Skeletal muscle is perhaps the most sensitive of all tissues to genetic mutations in genes that impact cellular attachment complexes or membrane repair capacity, in part because of the dynamic changes

**eLife digest** Muscle cells, also known as myofibers, need to be robust in order to withstand the physical stresses of contracting and relaxing. As a result, the cell surface membrane that surrounds myofibers is more strongly anchored to its surroundings than that of other cells. Muscular dystrophies are a group of muscle-wasting disorders that usually arise when this surface membrane becomes less stable. For example, mutations that affect a protein called dystrophin-glycoprotein or integrin protein complexes can cause muscular dystrophy since these proteins normally keep the membrane anchored and stable when the muscle contracts and relaxes.

When myofibers in mammals become injured, as is the case during muscular dystrophy, they produce more proteins called thrombospondins – with thrombospondin-4 being the most common. However, until now it was not clear what these proteins did in muscle cells.

Vanhoutte et al. hypothesized that thrombospondin-4 may protect injured myofibers and tested their theory by first deleting the gene for thrombospondin-4 from mutant mice that were predisposed to develop muscular dystrophy. This worsened the muscle wasting in the mutant mice, and furthermore, deleting the gene for thrombospondin-4 also caused otherwise normal mice to develop muscular dystrophy in their old age. Conversely, when Vanhoutte et al. artificially increased the levels of thrombospondin-4 in the myofibers, it protected the mice against muscular dystrophy. Additional experiments conducted in fruit flies demonstrated that the protective effects of thrombospondin are conserved or similar in insects too. Lastly, biochemical experiments in mouse and rat cells showed that thrombospondin-4 aids dystrophin-glycoproteins and integrins in getting to the cell surface membrane, increasing its stability.

Overall these findings provide a clearer picture of the molecular underpinnings of muscular dystrophies. In the future, more experiments will have to focus on exactly how thrombospondins stabilize and direct dystrophin-glycoproteins and integrins to the cell surface membrane.

in length that occurs in each myofiber during contraction (*Burr and Molkentin, 2015*; *Grounds et al., 2005*; *McNally and Pytel, 2007*).

Thrombospondins (Thbs) comprise a family of 5 genes in mammals that encode secreted matricellular proteins involved in diverse biologic processes (*Adams and Lawler, 2011* ; *Schellings et al., 2009*). The thrombospondin family consists of two subgroups based on their sequence conservation and oligomeric structure. Thbs3, Thbs4 and Thbs5 form pentamers and are the most similar to Thbs genes found in lower organisms (*Adams and Lawler, 2011*). Thbs1 and Thbs2 form trimers and have evolved additional domains such as a type 1 repeat important for transforming growth factor-β binding and a region that affects angiogenesis (*Adams and Lawler, 2011*; *Schellings et al., 2009*). *Drosophila* contains a single thrombospondin gene (Tsp) that forms pentamers, and when deficient causes developmental lethality due to disruption in muscle and tendon attachment within the body wall segments of the embryo (*Adams and Lawler, 2011*; *Subramanian et al., 2007*).

While traditionally characterized as a secreted ECM or matricellular protein over the past 3 decades, (*Adams and Lawler, 2011*; *Schellings et al., 2009*) Thbs can also function within the cell, and in some systems this appears to be their primary role (*Ambily et al., 2014*; *Baek et al., 2013*; *Brody et al., 2016*; *Duquette et al., 2014*; *Lynch et al., 2012*; *McKeown-Longo et al., 1984*; *Posey et al., 2014*). For example, Thbs4 was recently shown to have a critical cardioprotective function from within the endoplasmic reticulum (ER) where it mediates an adaptive ER stress response (*Brody et al., 2016*; *Lynch et al., 2012*). The traditional ER stress response involves sensing of calcium and unfolded or damaged proteins within the ER through the calcium binding chaperone protein BiP (GRP78), which binds/regulates at least 3 distinct stress response pathways initiated by either PKR-like ER kinase (PERK), inositol-requiring enzyme 1α (IRE1α) or activating transcription factor 6 (ATF6), each resident within the ER membrane (*Glembotski, 2007*; *Mori, 2009*). These 3 ER stress response mediators initiate a cascade of signaling that alters protein synthesis and other features of cellular adaptation to stress or protein unfolding and aggregation (*Glembotski, 2007*; *Mori, 2009*). Here, Thbs4 directly binds the ER luminal domain of ATF6α to promote its shuttling to

the Golgi and then nucleus, thereby inducing genes underlying adaptive aspects of the ER stress response (*Brody et al., 2016*; *Lynch et al., 2012*).

Thbs proteins move through the secretory pathway where they appear to facilitate secretion of ECM proteins or perhaps chaperone protein complexes to the cell membrane (*Adams and Lawler, 2011*). Once secreted, Thbs proteins transiently or permanently reside in the ECM where they interact with fibronectin, collagens and proteoglycans (*Adams and Lawler, 2011*; *Frolova et al., 2014*, *2012*; *Hauser et al., 1995*; *Södersten et al., 2006*). Thbs proteins are also recycled back into the cell through the low-density receptor-related protein (LRP) (*Wang et al., 2004*). One critical feature of the Thbs family is that each member is induced following injury events or in response to processes requiring tissue growth, healing and remodeling. Interestingly, Thbs4 is largely restricted to cardiac and skeletal muscle where its expression is induced with injury or disease (*Adams and Lawler, 2011*; *Chen et al., 2000*; *Frolova et al., 2014*, *2012*; *Hauser et al., 1995*; *Lynch et al., 2012*; *Schellings et al., 2009*; *Södersten et al., 2006*). In addition, markers of ER stress are upregulated during progression of skeletal muscle disease and MD (*Lavery et al., 2008*; *Moorwood and Barton, 2014*).

Here we observed that in response to MD in skeletal muscle, Thbs4 mRNA and protein are induced. Overexpression of Thbs4 in skeletal muscle of transgenic (Tg) mice protected against MD, while mice lacking Thbs4 (*Thbs4*$^{-/-}$) showed signs of spontaneous MD with aging. Mechanistically, Thbs4 directs a membrane attachment intracellular vesicular trafficking network that promotes greater stability of the DGC and integrin complexes at the sarcolemma of skeletal muscle fibers. This function of Thbs is conserved in *Drosophila* as overexpression of either mouse Thbs4 or *Drosophila* Tsp in muscle rescues MD that occurs in *Drosophila* deficient in its δ-sarcoglycan-related gene (*Allikian et al., 2007*).

## Results

### Thbs4 augments adaptive ER stress signaling in skeletal muscle and mitigates MD

In agreement with previous findings, Thbs4 RNA is induced in muscle biopsies from human patients with Becker MD, Duchenne MD, and limb-girdle MD (LGMD) (*Figure 1A*; *Figure 1—figure supplement 1A*) (*Chen et al., 2000*). We next turned to 2 different mouse models of MD. One due to deletion of the δ-sarcoglycan (*Sgcd*) gene to model LGMD2F, and a second due to defective dystrophin expression resulting from the *mdx* mutation that models Duchenne MD (*Durbeej and Campbell, 2002*). Thbs4 protein is induced in skeletal muscle of each mouse model at six weeks and three months of age, along with induction of an ER stress response associated with greater cleaved ATF6α-N (nuclear form) and increased total BiP levels (*Figure 1B,C*; *Figure 1—figure supplement 1B,C*).

To model the known increase in Thbs4 protein that occurs in dystrophic skeletal muscle we generated Tg mice with Thbs4 protein overexpression specific to skeletal muscle (*Figure 1D*). High levels of Thbs4 protein overexpression were observed in fast-twitch containing muscles such as the quadriceps, gastrocnemius, with intermediate levels in the diaphragm and very low levels in the soleus, while the heart lacked expression (*Figure 1D*). Tissue Immunofluorescent analysis revealed that Thbs4 protein was undetectable in uninjured skeletal muscle while the transgene produced abundant expression that co-localized with calreticulin to a vesicular network on the periphery of the myofibers of paraffin-embedded quadriceps and was also clearly inside of collagen I staining that marks the ECM of cryo-embedded quadriceps (*Figure 1E*). Furthermore, although Thbs4 protein localization appeared slightly different between paraffin- and cryo-embedded skeletal muscle of the *Sgcd*$^{-/-}$ mouse, induction of endogenous Thbs4 again showed localization within the vesicular network inside the myofibers, and only within limited regions outside of myofibers where fibrotic tissue deposition was prominent (*Figure 1E*).

In agreement with the above observations of Thbs subcellular localization, biochemical analysis revealed that Thbs4 was high in the type of glycosylation that typifies ER resident proteins, but it also contains glycosylation that is observed on proteins that transit through the Golgi in route to be deposited in the ECM (*Figure 1—figure supplement 2A,B*). Interestingly, in-depth in vitro analysis demonstrated that extracellular Thbs4 is rapidly internalized by both cultured C2C12 myoblasts and

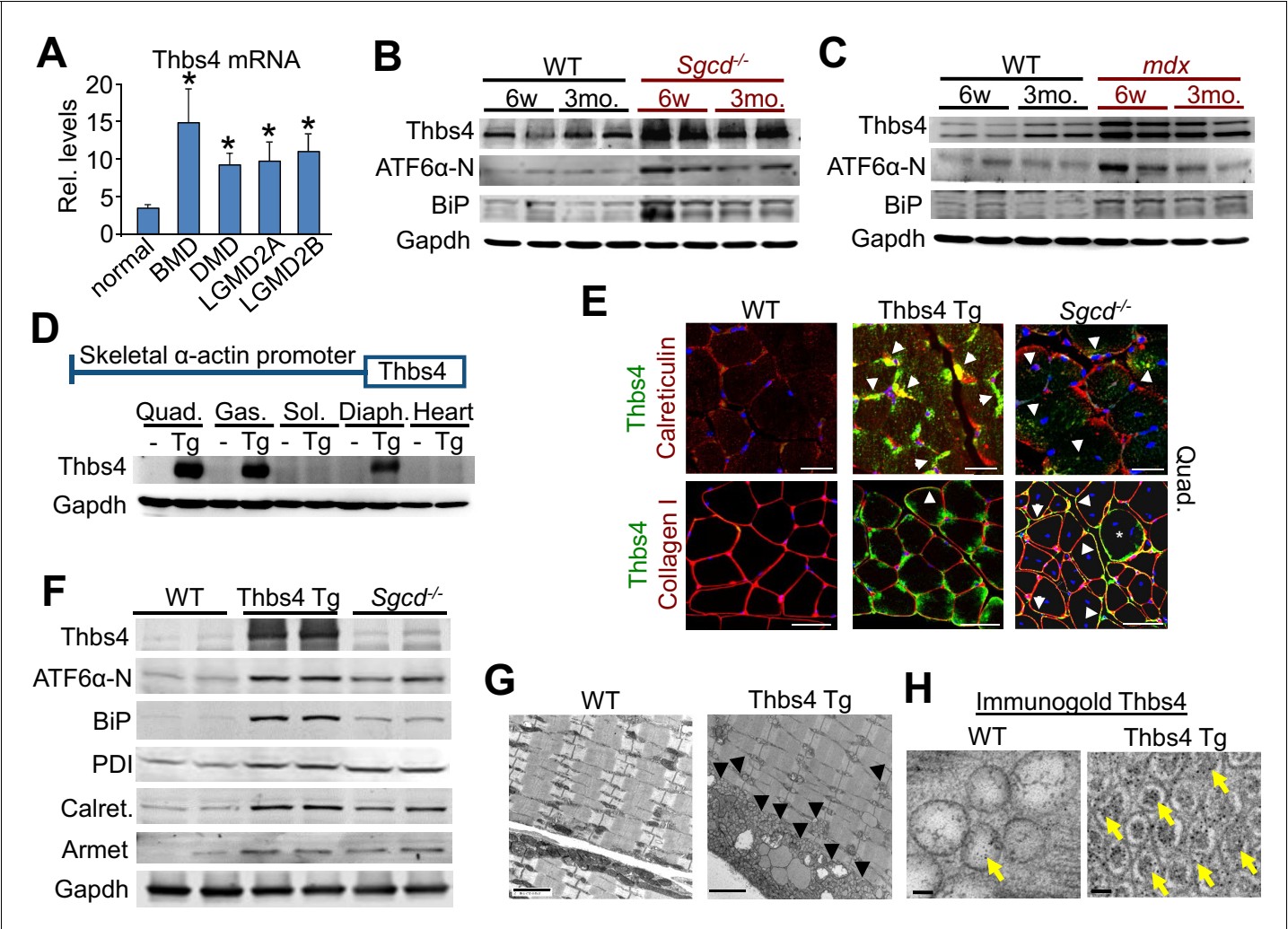

**Figure 1.** Thbs4 is induced in dystrophic skeletal muscle and its overexpression augments ER activity and vesicle content. (A) Thbs4 mRNA levels in human skeletal muscle biopsies from normal or patients with Becker MD (BMD; n = 5), Duchenne MD (DMD; n = 10) or 2 different types of limb-girdle MD (LGMD; n = 10 for both). *p<0.05 vs. normal (n = 18) by Student's *t* test. Data are presented as mean ± SEM. Full analysis including all 11 human muscle diseases is shown in *Figure 1—figure supplement 1A*. (B,C) Western blot for the expression of Thbs4, ATF6α-N (50 kDa, nuclear) and BiP in the quadriceps of WT, *Sgcd*−/− and *mdx* mice at six weeks (w) and three months (mo) of age (n = 4 biological replicates). (D) Schematic diagram showing the skeletal muscle-specific transgene to overexpress Thbs4 and (lower) Western blots for Thbs4 or gapdh control from WT and Tg mice at 6 w of age from Quad, quadriceps; Gas, gastrocnemius; Sol, soleus; Diaph, diaphragm; and heart (n = 2 biological replicates). (E) Upper micrographs represent co-immunofluorescent labeling of intracellular Thbs4 (green) with calreticulin (red) on paraffin embedded quadriceps (Quad.) of WT, Thbs4-Tg and *Sgcd*−/− mice at 3 mo of age (scale bar = 20 μm). Arrowheads indicate co-localization of Thbs4 with calreticulin in intracellular vesicles in the myofibers. Lower micrographs represent co- immunofluorescent labeling of Thbs4 (green) with collagen I (red) on cryo-embedded Quad of WT, Thbs4-Tg and *Sgcd*−/− mice at 3 mo of age (scale bar = 50 μm). Arrowheads indicate co-localization of Thbs4 with collagen I in the extracellular milieu; the star marks a myofiber with both intra- and extracellular Thbs4 labeling from a diseased muscle. Nuclei are visualized in blue. Representative images of 4 mice per genotype are shown. (F) Western blot analysis of Thbs4 and the ER-stress proteins ATF6α-N (50 kDa, nuclear), BiP, PDI, calreticulin (Calret.), and Armet in 6w old WT, Thbs4-Tg and *Sgcd*−/− quadriceps (n = 4 biological replicates). (G) Transmission electron microscopy in WT versus Thbs4 Tg quadriceps at 3 mo of age showing a massive expansion of intramyofibrillar and subsarcolemmal ER and associated vesicles with Thbs4 overexpression (arrowheads, scale bar = 2 μm). Representative images of 2 mice per genotype are shown. (H) Immunogold electron microscopy shows that Thbs4 (6 nm gold particles; yellow arrows) robustly localizes to the expanded sub-sarcolemmal vesicular compartment in Thbs4-Tg quadriceps, compared to endogenously expressed Thbs4 in WT quadriceps. Representative images of 2 mice are shown. Scale bar = 50 nm

The following figure supplements are available for figure 1:

**Figure supplement 1.** Thbs4 expression levels in human muscle diseases and quantitation of protein levels relative to Gapdh loading control of immunoblots shown in *Figure 1B, C and F*.

*Figure 1 continued on next page*

*Figure 1 continued*

**Figure supplement 2.** Analysis of Thbs4 glycosylation and vesicular expansion quantitation.
**Figure supplement 3.** Internalization of Thbs4 by cultured C2C12 myoblasts and myotubes.

myotubes and at least in the case of myoblasts is transported to rab7-positive late endosomes, potentially explaining why Thbs4 protein is low or undectable in the ECM of healthy Thbs4 Tg muscles (*Figure 1—figure supplement 3*). Similar to the results observed with collagen I, co-labeling with another ECM/matricellular protein periostin again showed that Thbs4 could co-localize to the ECM region in *Sgcd*$^{-/-}$ diseased myofibers, although under non-diseased conditions overexpressed Thbs4 was again only appreciably observed intracellularly under the sarcolemma within a peripheral vesicular pattern (*Figure 1E*; *Figure 1—figure supplement 2C*). Hence, although our observations do not exclude the possibility that non-muscle cells might also express Thbs4, our data collectively identify the myofiber as an important cellular source of Thbs4 expression, secretion and re-uptake.

Careful analysis of other markers of the ER compartment and ER stress showed that Thbs4 over-expression in skeletal muscle induced a profile very similar to diseased skeletal muscle in *Sgcd*$^{-/-}$ mice, with increased levels of nuclear ATF6α, BiP, protein disulfide isomerase (PDI), calreticulin and Armet, as compared to WT muscle (*Figure 1F*; *Figure 1—figure supplement 1D*). Remarkably, transmission electron microscopy and immunogold detection revealed that Thbs4 overexpression in skeletal muscle caused a dramatic induction of sub-sarcolemmal and intramyofibrillar ER and post-ER vesicles that contained Thbs4 protein (*Figure 1G,H*; *Figure 1—figure supplement 2D*). These Thbs4-dependent vesicles were highly uniform in size and more electron dense compared with similar vesicles in subsarcolemmal regions from WT muscle. Future studies will investigate the nature of these Thbs4-expanded vesicles and their composition based on known variables (*Malhotra and Erlmann, 2015*; *Paczkowski et al., 2015*).

To determine if Thbs4 induction in MD was adaptive or maladaptive we first crossed the Thbs4 Tg into both the *Sgcd*$^{-/-}$ and *mdx* backgrounds. Importantly, Thbs4 overexpression itself in skeletal muscle caused no histopathology or functional defects compared to WT mice at three and 12 months of age (*Figure 2A–E*; *Figure 2—figure supplement 1A–F*). More importantly, Thbs4 overexpression significantly reduced multiple histopathological hallmarks of dystrophic disease, including elevated serum creatine kinase (CK) levels, reduced myofiber degeneration/regeneration cycles as marked by reduced centrally nucleated myofibers, reduced fibrotic remodeling and less functional decline in skeletal muscle at both three and 12 months of age in both *Sgcd*$^{-/-}$ and *mdx* mice, compared with each dystrophic model alone (*Figure 2A–E*; *Figure 2—figure supplement 1A–F*, *Figure 2—figure supplement 2A–E*, *Figure 2—figure supplement 3A,B*).

The sarcolemma of dystrophic myofibers are weak and frequently rupture, which can be assessed in vivo by Evans blue dye (EBD) uptake into muscle fibers after systemic injection (*Goonasekera et al., 2011*; *Lapidos et al., 2004*). Here, the percentage of myofibers with ruptured membranes after forced treadmill running was significantly reduced in *Sgcd*$^{-/-}$ and *mdx* mice that contained the Thbs4 Tg, compared with *Sgcd*$^{-/-}$ and *mdx* mice alone, while no EBD uptake was observed in WT or Tg muscle (*Figure 2F,G*; *Figure 2—figure supplement 2F,G*). Ultrastructural analysis again showed that Thbs4 overexpression resulted in a dramatic induction in sub-sarcolemmal and intramyofibrillar vesicles in skeletal muscle of both *Sgcd*$^{-/-}$ and *mdx* mice (*Figure 2H*; *Figure 2—figure supplement 2H*).

This dramatic protection from MD observed in 2 mouse models of this disease with Tg-mediated overexpression of Thbs4 led us to investigate whether overexpression of Thbs4 would be sufficient to reduce acute dystrophic disease for the first time using a gene therapy-related approach to bypass possible developmental effects of overexpression. Hence, here we performed a study in *Sgcd*$^{-/-}$ mice with an adeno-associated virus serotype-9 (AAV9)-Thbs4 vector, which was injected into the gastrocnemius of three day-old neonates, followed by harvesting at six weeks of age to assess histopathology (*Figure 2—figure supplement 4A*). Littermates injected with an eGFP expressing AAV9 were used as a control. In agreement with our previous findings, this approach also resulted in abundant expression of either the control eGFP protein or a 5-fold increase in Thbs4 in the muscle

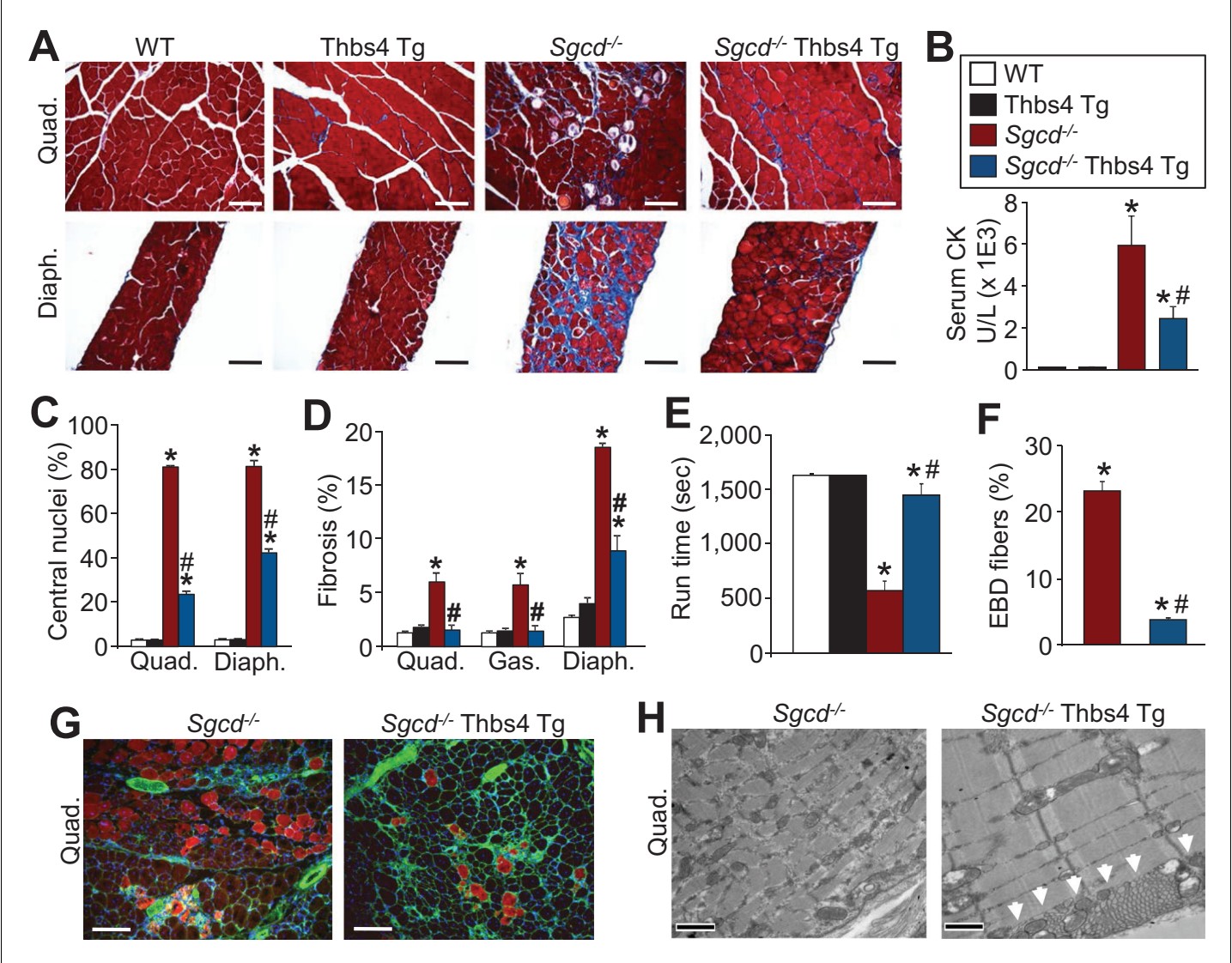

**Figure 2.** Thbs4 overexpression in skeletal muscle mitigates MD in mice. (A) Masson's trichrome stained sections of quadriceps (Quad.) and diaphragm (Diaph.) from WT, Thbs4-Tg, *Sgcd*-/- and *Sgcd*-/- Thbs4-Tg mice at 3 mo of age (Scale bar = 100 μm). Blue staining is fibrosis. (B) Quantitation of serum CK levels (units/liter) in indicated genotypes at 3 mo of age. *p<0.05 vs WT; #p<0.05 vs *Sgcd*-/- by one-way ANOVA with *post hoc* Tukey's test. N = 8 mice for WT, Thbs4 Tg and *Sgcd*-/- Thbs4 Tg and n = 7 mice for *Sgcd*-/-. The legend in this panel also refers to panels C–F. (C) Percentage of myofibers with centrally located nuclei in Quad and Diaph from H&E-stained histological sections at 3 mo of age. Representative images are shown in *Figure 2—figure supplement 3*. *p<0.05 vs WT; #p<0.05 vs *Sgcd*-/- by one-way ANOVA with *post hoc* Tukey's test. (D) Interstitial fibrosis analyzed in trichrome stained histological sections from Quad, gastrocnemius (Gas) and Diaph at three months of age. *p<0.05 vs WT; #p<0.05 vs *Sgcd*-/- by one-way ANOVA with *post hoc* Tukey's test. N = 6 mice for WT and Thbs4 Tg and n = 8 mice for *Sgcd*-/- and *Sgcd*-/-Thbs4 Tg in panel A, C and D. (E) Time to fatigue in seconds with forced downhill treadmill running at 3 mo of age in the indicated genotypes of mice. *p<0.05 vs WT; #p<0.05 vs *Sgcd*-/- by one-way ANOVA with *post hoc* Tukey's test. N = 6 mice per genotype. (F,G) Quantitation of total Evan's blue dye (EBD) positive fibers and representative immunofluorescent images of EBD uptake (red) in Quad of three month-old mice. Membranes of myofibers are shown in green. *p<0.05 vs WT; #p<0.05 vs *Sgcd*-/- by one-way ANOVA with *post hoc* Tukey's test. N = 5 mice per genotype in panel F. Scale bar = 100 μm. (H) Transmission electron microscopy of quadriceps muscle in *Sgcd*-/-Thbs4 Tg mice compared to *Sgcd*-/- mice at 3 mo of age. The arrowheads show subsarcolemmal vesicular expansion due to the Thbs4 Tg. Representative images of 2 mice per genotype are shown. Scale bar = 500 nm. All data are represented as mean ± SEM.

The following figure supplements are available for figure 2:

**Figure supplement 1.** Thbs4 overexpression mitigates muscular dystrophy in *Sgcd*-/- and *mdx* mice with aging to one year.

**Figure supplement 2.** Thbs4 overexpression mitigates MD in *mdx* mice at three months of age.

*Figure 2 continued on next page*

*Figure 2 continued*

**Figure supplement 3.** In depth imaging of H&E sections showing how Thbs4 overexpression reduces myofiber degeneration-regeneration (central nucleation) in both *Sgcd*[-/-] and *mdx* dystrophic quadriceps.

**Figure supplement 4.** Intramuscular AAV9-mediated overexpression of Thbs4 mitigates MD in *Sgcd*[-/-] mice.

of these mice (n = 3 mice per group, p<0.05 by Student's *t* test; *Figure 2—figure supplement 4B*). The results showed that AAV9-mediated Thbs4 overexpression significantly reduced central nucleation and fibrotic remodeling in *Sgcd*[-/-] mice with ongoing dystrophic disease (*Figure 2—figure supplement 4C–E*). Hence, overexpression of Thbs4 is protective and mitigates dystrophic disease even if initiated later in postnatal life.

## Loss of Thbs4 predisposes to MD

The induction of Thbs4 that normally occurs in skeletal muscle with dystrophic disease was further shown to be an adaptive and physiologic mechanism through the analysis of mice lacking the *Thbs4* gene. Here we crossed *Sgcd*[-/-] and *mdx* mice into the *Thbs4* null background and performed a full analysis of pathogenesis at three months of age. *Thbs4*[-/-] mice alone at three months of age showed minimal or no pathological changes in skeletal muscle, although combinatorial *Thbs4*[-/-] *Sgcd*[-/-] or *Thbs4*[-/-] *mdx* mice showed a significant worsening of MD, including greater histopathology, greater serum CK levels and reduced treadmill running performance, compared with single null *Sgcd* and *mdx* mice (*Figure 3A–C*; *Figure 3—figure supplement 1A–C*). *Sgcd*[-/-] *Thbs4*[-/-] mice also showed significantly greater EBD uptake in skeletal muscle after forced treadmill running versus *Sgcd*[-/-] mice alone (*Figure 3D,E*).

The observation that loss of the *Thbs4* gene makes dystrophic pathology significantly worse in both *Sgcd*[-/-] and *mdx* mice suggests that induction of this gene product plays an important protective role, and we reasoned that with aging loss of Thbs4 might eventually become pathologic to muscle given that low levels of continuous expression are present. Indeed, we observed that by six months and one year of age, *Thbs4*[-/-] mice showed a significant reduction in treadmill running capacity and increased serum CK levels compared to WT muscle (*Figure 3F,G*). Furthermore, by one year of age *Thbs4*[-/-] muscle had increased signs of ongoing myofiber degeneration/regeneration, as marked by centrally nucleated myofibers, as well as noticeable histopathological and ultrastructural changes and greater EBD uptake compared to WT control muscle (*Figure 3H–J*). Collectively, these results suggest that Thbs4 induction with dystrophic disease and low levels of expression during aging produce an adaptive physiologic response that protects skeletal muscle.

## Thbs4 directly impacts sarcolemma stability of myofibers

To directly evaluate the structural integrity of the sarcolemma we first employed a model of 3 successive lengthening-contraction injury cycles to the tibialis anterior (TA) muscle in a whole leg immobilization preparation (*Figure 4A*). Remarkably, overexpression of Thbs4 significantly protected against lengthening-contraction injuries over all 3 bouts compared to WT mice (*Figure 4B*). As anticipated, the TA from *Sgcd*[-/-] mice showed much greater loss of functional recovery compared to WT after lengthening-contraction injury, but the presence of the Thbs4 Tg provided significant protection, achieving a recovery response now similar to WT levels (*Figure 4B*). Moreover, loss of Thbs4 resulted in greater injury with all 3 cycles of lengthening-contraction injury, similar to *Sgcd*[-/-] (*Figure 4C*). Interestingly, the passive force of the muscle after simple stretching was greater with the Thbs4 Tg, while it was reduced in *Thbs4*[-/-] muscle, although the twitch force itself was not significantly different between any of the groups (*Figure 4—figure supplement 1A,B*). We also noticed that the tendons were weaker in *Thbs4*[-/-] TA muscle, which were more likely to rupture in the isolated lengthening-contraction injury assay (*Figure 4—figure supplement 1C*).

Individual myofibers were isolated from the flexor digitalis brevis (FDB) muscle and subjected to laser injury with subsequent measurement of FM1-43 fluorescent dye uptake as a direct measure of membrane stability and repair (*Figure 4D*). FDB myofibers from Thbs4 Tg mice showed less dye

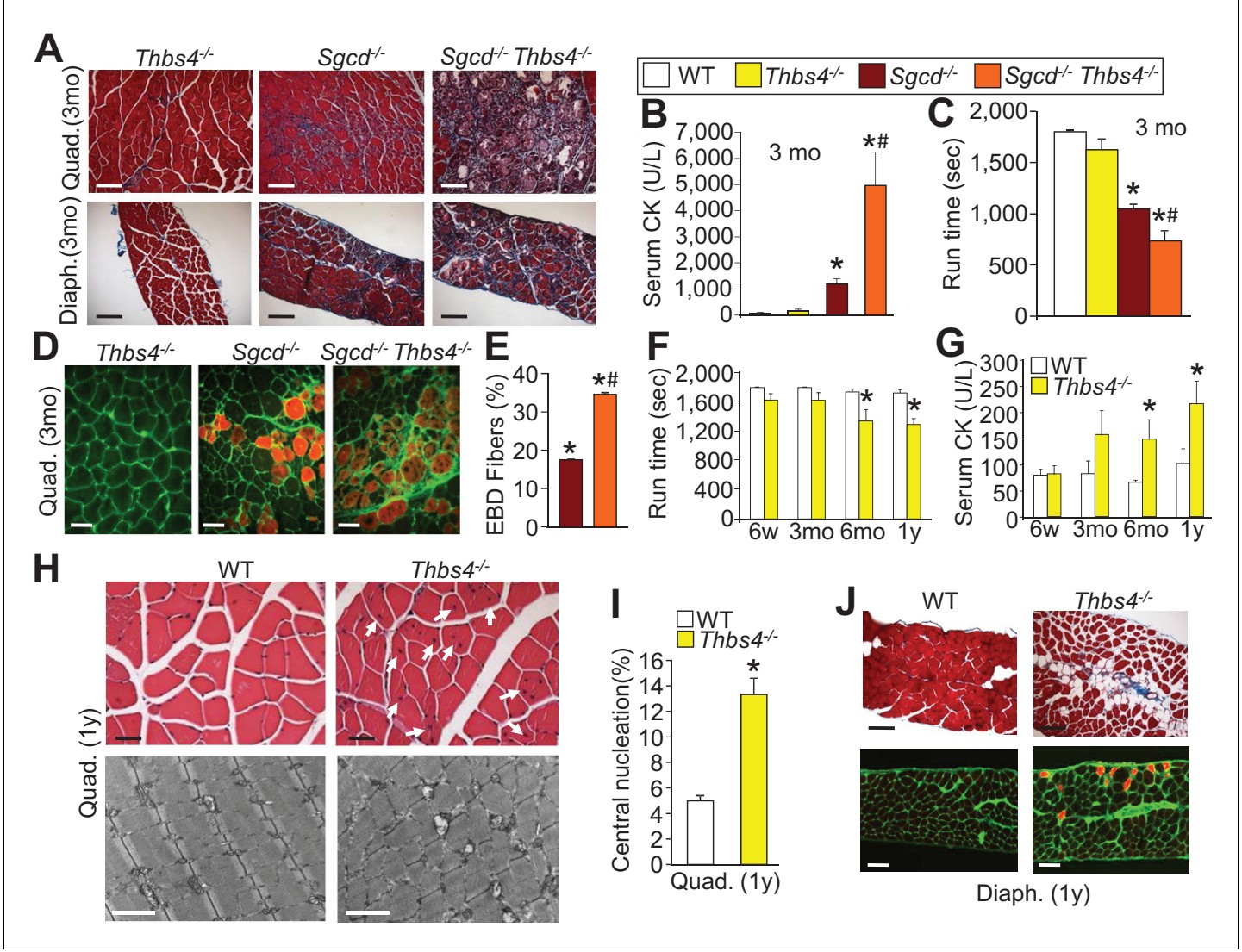

**Figure 3.** Loss of Thbs4 induces and exacerbates MD in mice. (A) Masson's trichrome stained histological sections of Quad and Diaph at three months (mo) of age in *Thbs4*-/-, *Sgcd*-/- and *Sgcd*-/-*Thbs4*-/- mice. Representative images of 6 mice for WT and Thbs4 Tg mice and 8 mice for *Sgcd*-/-and *Sgcd*-/-Thbs4 Tg. Scale bar = 100 μm. (B) Quantitation of serum CK (units/liter) in the indicated genotypes shown in the legend, at 3 mo of age. *p<0.05 vs. WT; #p<0.05 vs *Sgcd*-/- by one-way ANOVA with *post hoc* Tukey's test. N = 5 mice for WT and n = 6 mice for *Thbs4*-/-, *Sgcd*-/- and *Sgcd*-/-*Thbs4*-/-. (C) Time to fatigue in seconds with forced downhill treadmill running in mice at three months of age. *p<0.05 vs. WT; #p<0.05 vs. *Sgcd*-/- by one-way ANOVA with *post hoc* Tukey's test. N = 5 mice per genotype. The legend above panels **B** and **C** applies to the remainder of the figure. (D,E) Representative immunofluorescent images of EBD (red fluorescence) uptake and quantitation in Quad histological sections from three month-old mice. Membranes of myofibers are shown in green. *p<0.05 vs WT; #p<0.05 vs. *Sgcd*-/- by one-way ANOVA with *post hoc* Tukey's test. N = 5 and 6 mice for *Sgcd*-/- and *Sgcd*-/-*Thbs4*-/-, respectively. Scale bar = 40 μm. (F,G) Time to fatigue in seconds with forced downhill treadmill running and quantitation of serum CK levels in WT and *Thbs4*-/- mice at the indicated ages; abbreviations, y = year. n = 6 mice per genotype per age for panel **F**. For panel **G**, n = 7 mice per genotype at six weeks of age; n = 5 WT and 6 *Thbs4*-/-mice at three months of age, n = 5 mice per genotype at six months of age and n = 9 WT and 8 *Thbs4*-/-mice at one year of age. *p<0.05 vs WT at the same age by Student's t test. (H) H&E histological staining (upper) and transmission electron microscopy (lower) of tissue pathology in Quad at one year of age in WT and *Thbs4*-/-mice. The H&E scale bar = 25 μm. The electron microscopy scale bar = 2 μm. The arrows show myofibers with central nucleation due to loss of the *Thbs4* gene. Representative images of 6 mice per genotype for H&E staining and 2 mice per genotype for electron microscopy. (I) Percentage of myofibers with centrally located nuclei in *Thbs4*-/- compared to WT Quad at one year of age. *p<0.001 vs WT by Student's t test. n = 6 mice per genotype. (J) Masson's trichrome stained (upper) and EBD uptake (lower) histological images in WT and *Thbs4*-/-mice in the Diaph at one year of age. The EBD images show membranes in green and fibers with EBD uptake produce red fluorescence. Representative images of 6 mice per genotype studied. Scale bars = 50 μm. All data are represented as mean ± SEM.

The following figure supplement is available for figure 3:

*Figure 3 continued*

**Figure supplement 1.** *Thbs4*[-/-] mice show enhanced MD pathology in the *mdx* background.

uptake compared with WT myofibers, suggesting that Thbs4 overexpression was inherently protective to the membrane (***Figure 4D,E***). As expected, FDB myofibers from *Sgcd*[-/-] mice showed greater dye uptake suggesting greater injury with less efficient repair, while the presence of the Thbs4 Tg was protective in *Sgcd*[-/-] myofibers, bringing dye uptake levels back to WT (***Figure 4D,E***). FDB myofibers from *Thbs4*[-/-] mice also showed greater dye uptake compared with WT myofibers after laser injury, while double *Thbs4*[-/-] *Sgcd*[-/-] myofibers displayed an even greater injury response than either *Thbs4* or *Sgcd* single deletions alone (***Figure 4F***). Taken together these results indicate that Thbs4 overexpression provides greater stability to the sarcolemma, while its loss renders the sarcolemma less stable. More importantly, since this assay uses isolated myofibers devoid of ECM attachments, it suggests that part of Thbs4-dependent protection occurs from within the myofiber.

## Thbs4 augments intracellular trafficking through ATF6α

To investigate a molecular mechanism whereby Thbs4 might directly regulate the stability of the sarcolemma in skeletal muscle we examined intracellular vesicular trafficking and membrane attachment complex formation. Two important clues that directed our investigation were the known augmentation in the adaptive ER stress response pathway and the dramatic increase in sub-sarcolemmal vesicles observed in skeletal muscle with Thbs4 overexpression. Hence, we instituted two in vitro assays to assess ER-to-Golgi and Golgi-to-sarcolemma vesicular trafficking in response to Thbs4 activity in primary neonatal rat ventricular myocytes (***Figure 5***; ***Figure 5—figure supplement 1A,B***, technical issues prevented such studies in myotubes or myofibers). To assess ER-to-Golgi trafficking a red fluorescent protein (RFP)-labeled Golgi resident enzyme, GalNacT2-RFP, was used with and without Thbs4 overexpression by fluorescent recovery after photobleaching (FRAP, ***Figure 5—figure supplement 1A***). In parallel, ATF6α activity was also modulated since Thbs4 is known to directly regulate this ER-stress transcription factor (***Brody et al., 2016***; ***Lynch et al., 2012***). Here, Thbs4 overexpression significantly accelerated ER-to-Golgi vesicular trafficking, which was fully inhibited by co-overexpression of a dominant negative (dn) ATF6α construct (***Figure 5A***). Similarly, Golgi-to-sarcolemmal trafficking rates, measured with VSVG-enhanced green flourescent protein (eGFP) after inverse (i) FRAP, were significantly accelerated upon Thbs4 overexpression, which was again inhibited with ATF6α-dn (***Figure 5B***; ***Figure 5—figure supplement 1B***). Furthermore, accelerated trafficking with Thbs4 was mimicked by overexpression of a constitutively nuclear (cn) ATF6α construct (***Figure 5C,D***).

To examine this effect of Thbs4 in greater molecular detail we also employed 2 domain-specific Thbs4 constructs and a related oligomeric glycoprotein Nell2 (***Figure 5E–J***) (***Brody et al., 2016***; ***Kuroda et al., 1999***; ***Lynch et al., 2012***). Like Thbs4, Nell2 contains an N-terminal laminin-G like (LamG) domain and an epidermal growth factor (EGF)-like repeat domain but lacks the ATF6α interacting Type III repeat (T3R) and TSP-C domains from Thbs4. Importantly, unlike full-length Thbs4, overexpression of Nell2 or the LamG domain of Thbs4 did not increase ER-to-Golgi or post-Golgi trafficking rates (***Figure 5E,F,I,J***). However, overexpression of just the T3R domain of Thbs4, which functions as the ATF6α interacting region, was sufficient to accelerate ER to Golgi and post-Golgi trafficking (***Figure 5G,H***). Thbs4 overexpression in skeletal muscle also augmented the levels of trafficking regulatory proteins such as Sar1, Rab24, Rab6, Rab3 and Rab8, which control ER-to-Golgi and post-Golgi vesicular trafficking (***Figure 5—figure supplement 1C***) (***Brandizzi and Barlowe, 2013***; ***Stenmark, 2009***). Collectively, these results indicate that Thbs4 accelerates intracellular vesicular trafficking in an ATF6α-dependent manner, thereby suggesting at least one molecular mechanism whereby Thbs4 might enhance membrane stability through greater fluxing of vesicles to the sarcolemma.

## Thbs4 stabilizes sarcolemmal attachment complexes

To more definitively investigate if greater vesicular trafficking rates associated with Thbs4 activity might augment the residency of membrane attachment complex proteins, we generated membrane-

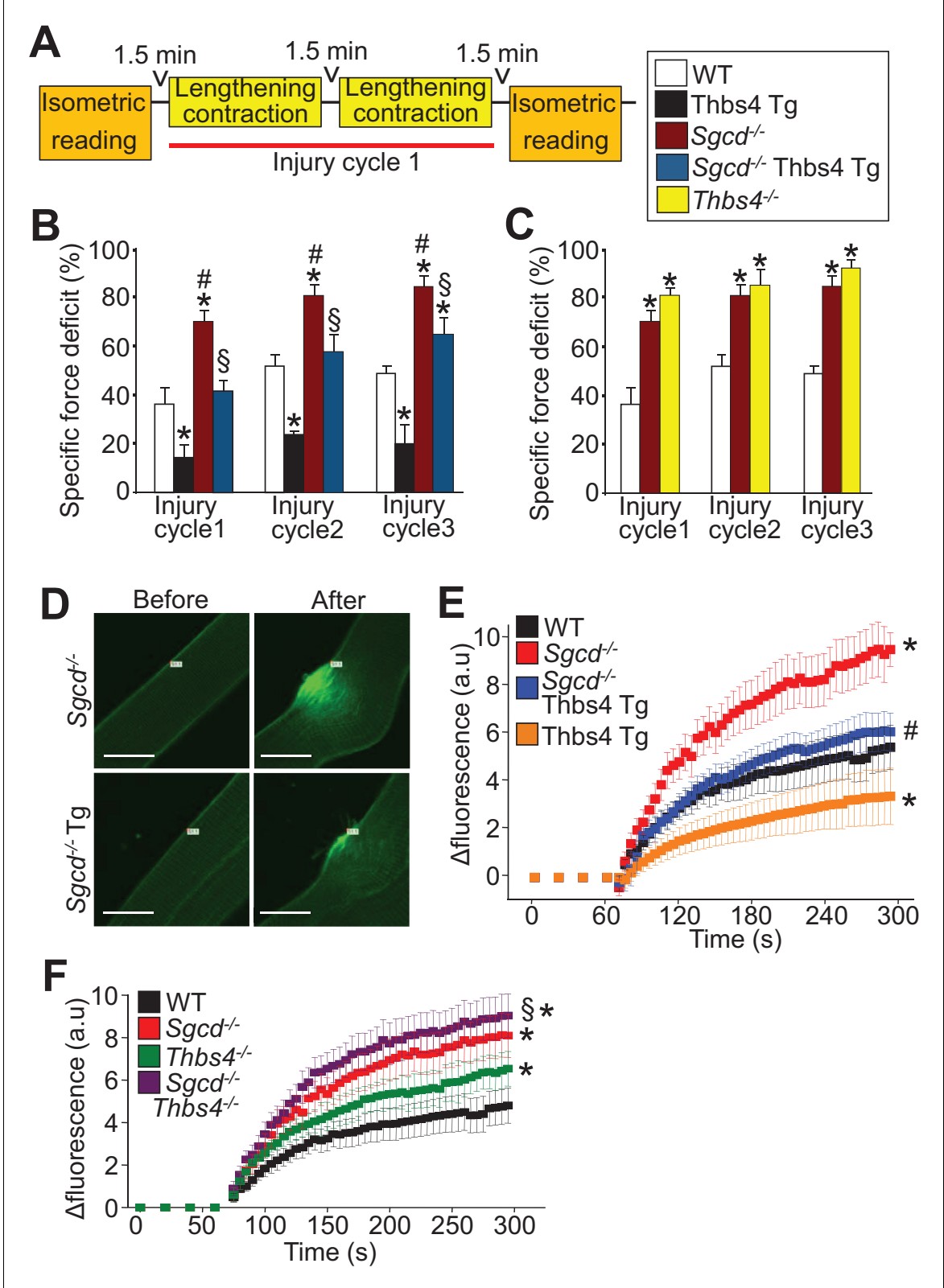

**Figure 4.** Thbs4 regulates skeletal muscle sarcolemma stability. (**A**) Schematic representing the first of 3 consecutive lengthening contraction-induced muscle injury cycles using an in situ tibialis anterior (TA) muscle preparation. Briefly, an isometric contraction was performed to determine baseline force generation, followed by 2 consecutive eccentric contractions and finally another isometric contraction (= injury cycle 1). The force deficit shown in panels B and C was calculated between the first and second isometric contraction in between the two lengthening-contractions, which was repeated 3

*Figure 4 continued on next page*

*Figure 4 continued*

cycles total. (B,C) Reduction in isometric force generation as a percentage of baseline force after each lengthening contraction injury cycle in the indicated genotypes of mice shown in the legend. *p<0.05 vs WT; #p<0.05 vs WT and Thbs4-Tg; §P<0.05 vs *Sgcd*[-/-]and Thbs4-Tg by one-way ANOVA with *post hoc* Tukey's test. n = 6 mice for WT, Thbs4 Tg and *Sgcd*[-/-]Thbs4 Tg and n = 10 mice for *Sgcd*[-/-] for panel B. n = 6 mice for WT, n = 10 mice for *Sgcd*[-/-] and n = 5 mice for *Thbs4*[-/-] for panel C. (D) Representative images before and after laser injury and influx of FM1-43 dye (green fluorescence) in FDB myofibers in the presence of 1.25 mM $Ca^{2+}$ isolated from indicated genotypes. The white tag in each image is the position for the laser injury. Scale bars = 10 μm. (E,F) Quantitative time course in seconds of FM1-43 fluorescent dye entry in FDB myofibers from the indicated genotypes of mice in the presence of 1.25 mM $Ca^{2+}$. Laser injury occurred at 60 s. n = 6 fibers per animal from 3 animals per genotype for panels D–F; *p<0.05 vs WT; #p<0.05 vs *Sgcd*[-/-]; §p<0.05 vs *Thbs4*[-/-] by one-way ANOVA with *post hoc* Tukey's test. Data points for *Sgcd*[-/-]in panel E and F were derived from a single set of experiments. All data are represented as mean ± SEM.

The following figure supplement is available for figure 4:

**Figure supplement 1.** Thbs4 alters the mechanical and structural properties of muscle and tendons.

specific protein preparations for Western blotting, as well as performed immunohistochemistry on skeletal muscle for direct visualization of the membranes in cross-section. As previously observed, loss of δ-sarcoglycan in skeletal muscle of *Sgcd*[-/-] mice resulted in the near complete absence of the other sarcoglycans at the membrane (as they all form a complex; *Figure 6A*) (*Durbeej and Campbell, 2002*). Interestingly, Thbs4 Tg alone displayed increased sarcolemmal levels of β-dystroglycan, dystrophin and β1D-integrin (*Figure 6A*). More importantly, the Thbs4 Tg in the *Sgcd* deficient background resulted in greater membrane localization of α-, β-, and γ-sarcoglycan and β-dystroglycan, as well as in dystrophin, utrophin and β1D-integrin (*Figure 6A*). More quantitative assessment of this effect by Western blotting of membrane-specific protein extracts showed that even the Thbs4 Tg alone gave increased membrane levels of utrophin, α- and β-dystroglycan, as well as β1D-, α7 and α5-integrins compared to WT controls (*Figure 6B*; *Figure 6—figure supplement 1A,B*; red boxes), which likely explains our earlier observations whereby Thbs4 Tg skeletal muscle was protected from lengthening-contraction injury and laser injury in isolated myofibers versus WT. The Thbs4 Tg also augmented the membrane residency of these same proteins in the *Sgcd*[-/-] and *mdx* background, as well as increased membrane levels of α-sarcoglycan and β-sarcoglycan in skeletal muscle of *Sgcd*[-/-] mice (*Figure 6B*; *Figure 6—figure supplement 1A,B*, for replicate samples). Importantly, previous work has shown that overexpression of DGC components augment the assembly of the entire complex and are inherently protective to MD (*Allikian et al., 2004*; *Grounds et al., 2005*; *Gumerson and Michele, 2011*; *Tinsley et al., 1998*). Furthermore, we observed that expression of β1- and α7-integrin from total cytoplasmic protein extracts was not increased, suggesting that Thbs4 overexpression directly augmented membrane trafficking and localization of these critical attachment proteins to the surface (*Figure 6B*, lower panel). Finally, we also observed that loss of Thbs4 in skeletal muscle partially reduced membrane residency of α- and β-dystroglycan, β1D- and α7-integrins (*Figure 6C*, burgundy boxes). Collectively, these observations solidify a mechanism whereby Thbs4 regulates membrane stability in skeletal muscle by augmenting the trafficking of membrane attachment protein complexes to the sarcolemma.

As previously reported with other Thbs proteins (*Adams and Lawler, 2011*), we observed that Thbs4 could directly bind intracellular β1D-integrin, but not α7-integrin, in primary neonatal rat cardiomyocytes (*Figure 6—figure supplement 1C*). In addition, both Thbs4 and α5-integrin localized to β1D-integrin-positive intracellular vesicles in WT, Thbs4 Tg and *Sgcd*[-/-] quadriceps muscle, placing Thbs4 within the same vesicles and intracellular compartment as the attachment proteins themselves (*Figure 6—figure supplement 1D,E*).

## ATF6α drives vesicular expansion in skeletal muscle but does not protect against MD

Given the results presented to this point, we hypothesized that ATF6α induced ER and post-ER vesicular expansion was responsible for increased intracellular trafficking of membrane attachment protein complexes to the sarcolemma and hence, protection from MD. Indeed, ATF6α was previously shown to expand the ER and post-ER vesicular compartment when activated or overexpressed (*Brody et al., 2016*; *Lynch et al., 2012*). Thus, to directly test this hypothesis we generated skeletal

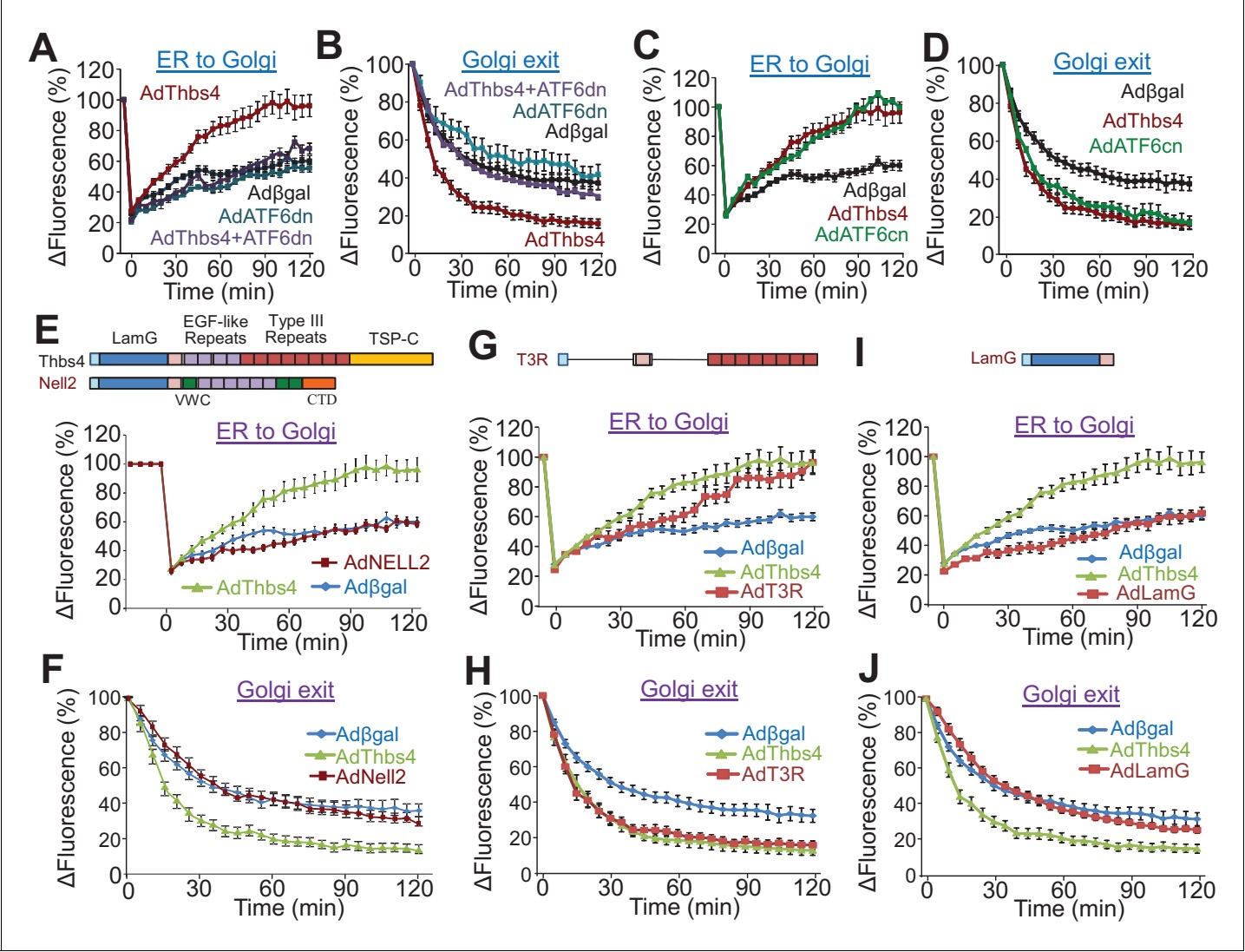

**Figure 5.** Thbs4 enhances intracellular vesicular trafficking through its ATF6α interacting region. (A–D) Time course of GalNac-T2-RFP or VSVG-eGFP fluorescence changes in cultured neonatal ventricular myocytes infected with the indicated adenoviruses to overexpress Thbs4 (maroon line, n = 18), a dominant negative (dn) ATF6α (turquoise line, n = 8) or in combination with Thbs4 (purple line, n = 13), a constitutively nuclear (cn) ATF6α (green line, n = 12), or βgal expressing control (black line, n = 14). Change in fluorescence was after FRAP or iFRAP to measure ER-to-Golgi or Golgi to the membrane trafficking, respectively. (E,G,I) Quantitative time course of GalNac-T2-RFP recovery in the Golgi network after FRAP to measure ER to Golgi vesicular trafficking in primary neonatal rat ventricular myocytes infected with adenoviral Thbs4 (green line, n = 18), Nell2 (red line, n = 9), the Type III repeat (AdT3R; red line, n = 8) domain of Thbs4, the N-terminal Laminin G (AdLamG, red line, n = 7) domain of Thbs4, or βgal control (blue line, n = 14). Shown above these graphs is a schematic diagram depicting the domain structure of Thbs4, Nell2 and T3R or LamG of Thbs4. All data are represented as mean ± SEM. (F,H,J) Quantitative time course of loss of VSVG-eGFP fluorescence in the Golgi after iFRAP as a measurement for Golgi-to-membrane (Golgi exit) vesicular trafficking in primary neonatal rat ventricular myocytes infected with adenoviral Thbs4 (green line, n = 18), Nell2 (red line, n = 8), the Type III repeat (AdT3R; red line, n = 8) domain of Thbs4, the N-terminal Laminin G (AdLamG; red line, n = 7), or βgal control (blue line, n = 14). All data are represented as mean ± SEM. Each cell imaged per experimental condition represents an independent experiment.

The following figure supplement is available for figure 5:

**Figure supplement 1.** Thbs4 enhances intracellular vesicular trafficking.

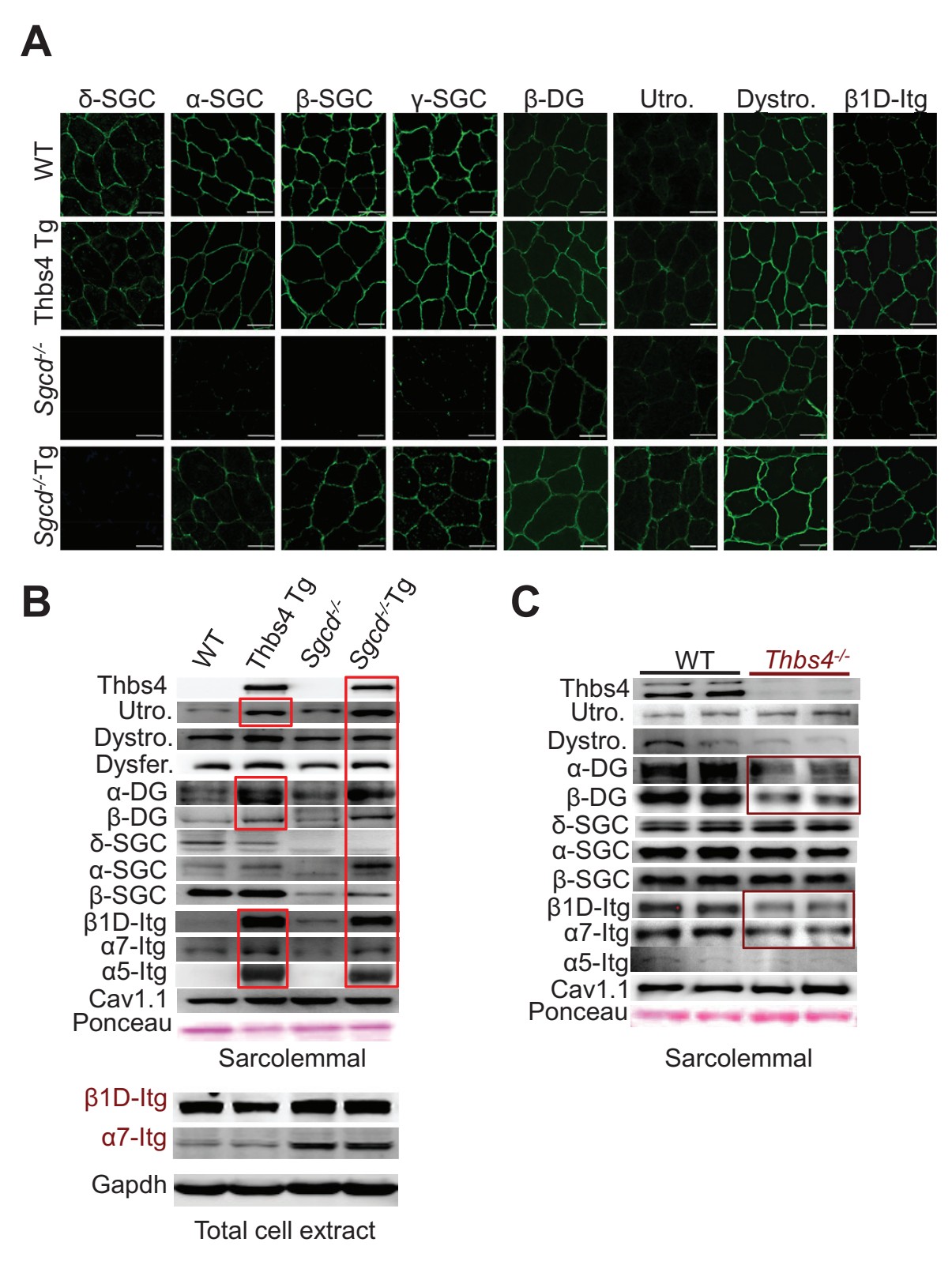

**Figure 6.** Thbs4 enhances stabilizing proteins at the sarcolemma. (**A**) Immunofluoresence (green) detection of δ-, α-, β-, and γ-sarcoclycan (SGC), β-dystroglycan (β-DG), utrophin (Utro.), dystrophin (Dystro.) and β1D-integrin in littermates of three month-old WT, Thbs4-Tg, *Sgcd⁻/⁻* and *Sgcd⁻/⁻*Thbs4-Tg quadriceps. Representative images of 4 mice per genotype are shown. Scale bar = 25 μm. (**B**) Representative Western blots of sarcolemmal protein extracts (upper) or total cytoplasmic protein extracts (lower) from the quadriceps of the indicated groups of mice for the indicated proteins (n = 4–5

*Figure 6 continued on next page*

*Figure 6 continued*

biological replicates). Abbreviations: Utro, utrophin; Dystro, dystrophin; Dysfer, dysferlin; α-DG, α-dystroglycan; β-DG, β-dystroglycan; δ-SCG, δ-sarcoglycan; α-SCG, α-sarcoglycan; β-SGC, β-sarcoglycan; β1D-, α7- and α5- integrin. The red boxes show increased protein levels. Also see *Figure 6—figure supplement 1* for replicates. (C) Representative immunoblotting for structural components of the DGC- and integrin-associated protein complexes in sarcolemmal preparations from *Thbs4^{-/-}* and WT quadriceps at four months of age (n = 4 biological replicates). The burgundy-boxed areas show reduced protein levels. Ponceau staining of a nonspecific band and dihydropyridine receptor α1 (Cav1.1) were used as loading controls for sarcolemmal protein extracts; Gapdh was used as loading control for total cell protein extracts.

The following figure supplement is available for figure 6:

**Figure supplement 1.** Thbs4 enhances stabilizing proteins at the sarcolemma and directly interacts with integrins.

muscle-specific Tg mice overexpressing ATF6α, which showed high levels of nuclear ATF6α protein and induction of BiP, PDI and calreticulin as compared to WT levels, all without influencing Thbs4 protein expression levels (*Figure 7A,B*; *Figure 7—figure supplement 1*). Moreover, except for PDI we noted significantly higher levels of these ATF6α-dependent ER stress responsive factors in ATF6α Tg quadriceps as compared to our Thbs4 Tg (*Figure 7A,B*; *Figure 7—figure supplement 1*). ATF6α skeletal muscle-specific Tg mice appeared overtly normal and showed no histopathology of skeletal muscles (*Figure 7C*; *Figure 7—figure supplement 2A,B*). Similar to Thbs4 Tg mice, ultra-structural analysis of skeletal muscle from ATF6α Tg mice showed a remarkable expansion of ER and the sub-sarcolemmal vesicular compartment, although these ATF6α-dependent vesicles appeared less dense in comparison to those observed in Thbs4 Tg mice (*Figure 7D* versus *Figure 1G*).

Next, ATF6α mice were crossed with both *Sgcd^{-/-}* and *mdx* mice to directly examine the hypothesis that the adaptive ER stress response and sub-sarcolemmal expansion of vesicles induced by ATF6α was a protective mechanism underlying Thbs4 action. However, ATF6α overexpression in the *Sgcd^{-/-}* or *mdx* dystrophic background provided no protection whatsoever (*Figure 7E–J*; *Figure 7—figure supplement 2A–D*). There was no reduction in histopathology or serum CK levels or membrane rupture as assessed with EBD, nor was treadmill running improved by the ATF6α Tg in either the *Sgcd^{-/-}* or *mdx* backgrounds. More importantly, ATF6α overexpression did not increase the sarcolemmal localization of any of the membrane attachment proteins observed with Thbs4 overexpression in skeletal muscle (*Figure 7K*). Thus, while ATF6α overexpression activated an adaptive ER stress response in skeletal muscle with a dramatic induction of intracellular and sub-sarcolemmal vesicles to an even higher level than observed in our Thbs4 overexpressing mice, it did not augment the membrane residency of membrane stabilizing proteins in muscle. Hence, our data indicate that ATF6α is only one part of a more integrated mechanism whereby Thbs4 regulates membrane stability of skeletal muscle.

Taken together, our data so far indicate that increased levels of Thbs4 itself and its trafficking through the secretory pathway are essential to increase membrane attachment protein complexes at the sarcolemma. To test this hypothesis, we took advantage of a previously established adenoviral construct encoding a Thbs4 calcium-binding containing mutant that is retained in the ER (Ad-Thbs4-mCa$^{2+}$) (*Brody et al., 2016*). Importantly, the mutant still induces an ATF6α mediated ER-stress response, both in neonatal rat cardiomyocytes and when expressed in the gastrocnemius muscle of early postnatal rat pups (*Brody et al., 2016*). Utilizing an identical in vivo approach with adenoviral gene transfer into the gastrocnemius of early neonatal rat pups, we compared βgal control with full-length Thbs4 versus the Thbs4-mCa$^{2+}$ for effects on β1 Integrin membrane levels. The data showed that only the secretion competent full-length Thbs4, but not the full-length ER-retained Thbs4 mutant, promoted greater β1 integrin membrane occupancy (*Figure 7—figure supplement 3*). Hence, Thbs4 must move through the secretory pathway to chaperone at least β1 integrin to the sarcolemma.

## Conservation of the Thbs membrane stability mechanism in *Drosophila*

Our results in mice were reminiscent of data from *Drosophila*, which have a single Tsp gene that when deficient causes embryonic lethality due to ruptures in tendon/muscle attachments (*Subramanian et al., 2007*). Moreover, Tsp in *Drosophila* was also shown to interact with αPS2/βPS integrin (*Chanana et al., 2007*; *Subramanian et al., 2007*). Thus to investigate a potential

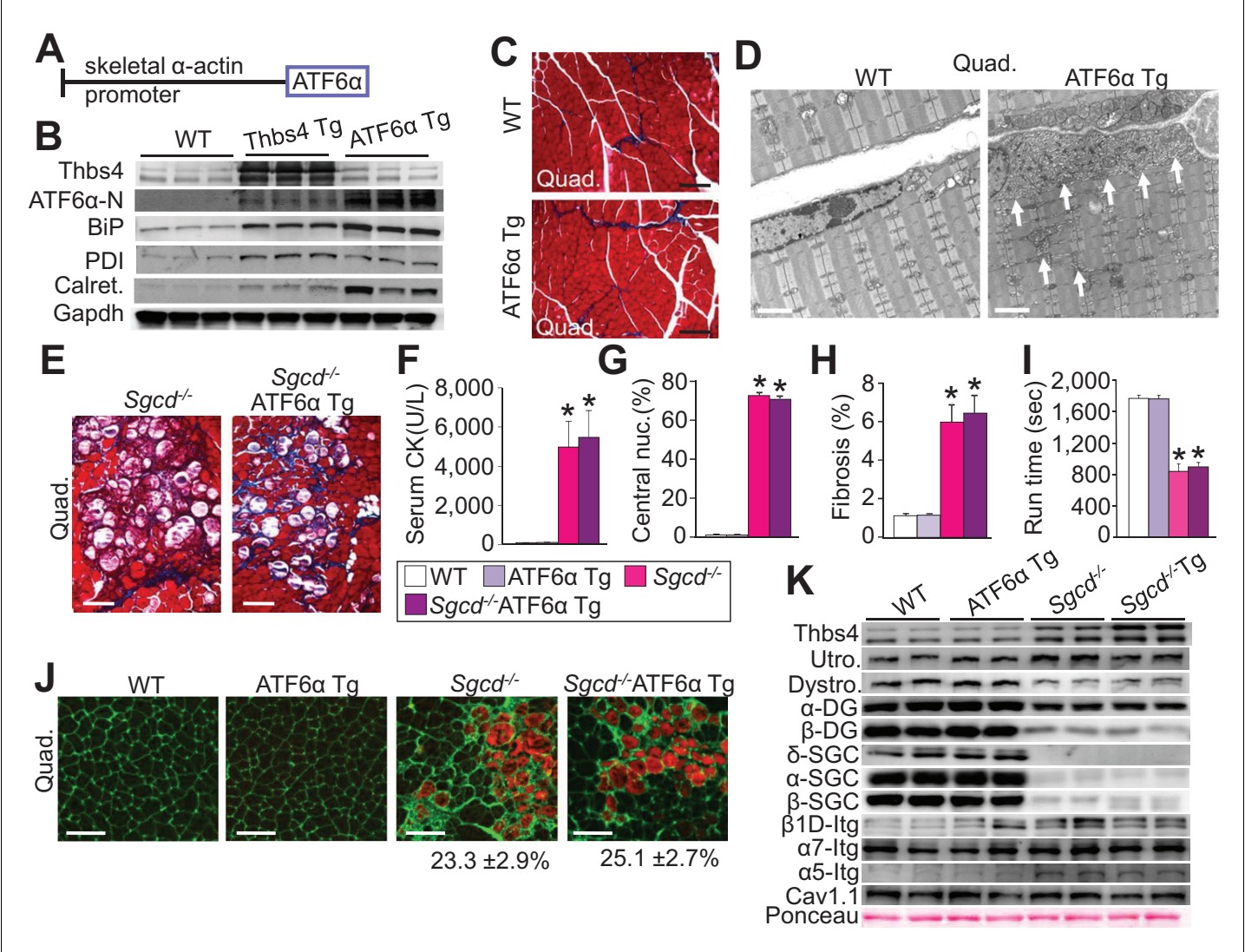

**Figure 7.** Skeletal muscle specific ATF6α overexpression drives ER stress and intracellular vesicular expansion, but not protection against MD. (**A**) Schematic diagram of the transgene (Tg) used to overexpress ATF6α in skeletal muscle. (**B**) Western blot analysis for Thbs4, ATF6α, BiP, PDI and calreticulin (Calret.) expression in quadriceps (Quad) from WT, Thbs4 Tg and ATF6α Tg mice at six weeks of age. Gapdh is a processing and loading control (n = 6 biological replicates). (**C**) Masson's trichrome-stained histological sections from Quad of WT and ATF6α-Tg littermates at six weeks of age. Representative images of 5 mice per genotype are shown. Scale bar = 100 µm. (**D**) Transmission electron micrographs in Quad from WT and ATF6α-Tg mice at six weeks of age. The white arrows show dramatic expansion of ER and associated vesicles throughout the cell and especially in the sub-sarcolemmal region. Representative images of 2 mice per genotype are shown. Scale bar = 2 µm. (**E**) Masson's trichrome stained histological sections of Quad from *Sgcd*[-/-] and *Sgcd*[-/-] ATF6α Tg mice at six weeks of age. Representative images of 5 mice per genotype are shown. Scale bar = 100 µm. (**F**) Quantitation of serum CK levels (units/liter) in the indicated genotypes of mice shown in the legend below the graph at six weeks of age. n = 10 mice for WT and n = 8 mice for the remaining genotypes. *p<0.05 versus WT by one-way ANOVA with *post hoc* Tukey's test. (**G,H**) Histological analysis of the Quad showing percentage of myofibers with centrally located nuclei (n = 5 mice for WT and ATF6α Tg, and n = 6 mice for *Sgcd*[-/-] and *Sgcd*[-/-]ATF6α Tg) and interstitial fibrosis (n = 5 mice per genotype) at six weeks of age in WT, ATF6α Tg, *Sgcd*[-/-] and *Sgcd*[-/-]ATF6α Tg mice. *p<0.05 versus WT by one-way ANOVA with *post hoc* Tukey's test. (**I**) Time to fatigue in seconds with forced downhill treadmill running in the indicated genotypes of mice shown in the legend. n = 6 mice per genotype. *p<0.05 versus WT by one-way ANOVA with *post hoc* Tukey's test. (**J**) Representative immunofluorescent images of EBD (red) uptake in myofibers in the Quad of six week-old mice of the indicated genotypes. Membranes of myofibers are shown in green. Scale bars = 75 µm. Percent EBD-positive myofibers is indicated. Six mice per genotype were analyzed for EDB uptake. (**K**) Western for structural components of the DGC and integrin-associated protein complexes in sarcolemmal protein preparations from Quad of WT, ATF6α Tg, *Sgcd*[-/-] and *Sgcd*[-/-]ATF6α Tg littermates at six weeks of age. Ponceau staining of a nonspecific band and dihydropyridine receptor α1 (Cav1.1) were used as loading controls (n = 3 biological replicates). Abbreviations: Utro, utrophin; Dystro, dystrophin; α-DG, α-dystroglycan; β-DG, β-dystroglycan; δ-SCG, δ-sarcoglycan; α-SCG, α-sarcoglycan; β-SGC, β-sarcoglycan; β1D-, α7- and α5-Itg (integrin). All data are represented as mean ± SEM.

*Figure 7 continued on next page*

*Figure 7 continued*

The following figure supplements are available for figure 7:

**Figure supplement 1.** Relative protein levels for immunoblots shown in *Figure 7B*.

**Figure supplement 2.** ATF6α skeletal muscle-specific Tg mice are not protected from MD in the *mdx* genetic background.

**Figure supplement 3.** Complete membrane trafficking of Thbs4 is essential for its membrane stabilizing function.

conservation of Thbs4's function, we generated Tg *Drosophila* in which the mouse Thbs4 cDNA or the *Drosophila* Tsp cDNA were driven with a muscle-specific myocyte enhancer factor 2 (MEF2) regulatory region to achieve overexpression of either protein in *Drosophila* muscle. These lines were subsequently crossed with a *Drosophila* model of MD that lacks the δ-sarcoglycan-like gene (Sgcd$^{840}$), a line that is marked by muscle disease with shortened life-span, loss of muscle function and overt rupture of the muscles due to structural weakness (*Allikian et al., 2007*). Remarkably, both Thbs4 and Tsp rescued the reduced life span in the Sgcd$^{840}$ MD *Drosophila* line, and normalized muscle function as assessed with a negative geotaxis assay (*Figure 8A,B*; *Figure 8—figure supplement 1A,B*). The rupture of the dorsal median indirect flight muscles and their loss of proper ectoskeletal attachment were also rescued by muscle-specific overexpression of either mouse Thbs4 or *Drosophila* Tsp (*Figure 8C*; *Figure 8—figure supplement 1C*).

Mechanistically, Thbs4 or Tsp overexpression was exclusively restricted to a vesicular compartment within the indirect flight muscles of these overexpressing Tg lines, but not outside the myofibers (*Figure 8D,H*; *Figure 8—figure supplement 1D*). More provocatively, both mouse Thbs4 and *Drosophila* Tsp overexpression produced noticeably greater levels of membrane βPS integrin localization in *Drosophila* muscle (*Figure 8E,F*; *Figure 8—figure supplement 1E,F*). Ultrastructural analysis revealed a remarkable rescue of the sarcomeric tears in Sgcd$^{840}$*Drosophila* with the mouse Thbs4 or *Drosophila* Tsp Tg, although without inducing ER-stress nor ER and post-ER vesicular expansion in these muscles (*Figure 8F,G*; *Figure 8—figure supplement 1G–I*). Collectively, these results indicate that Thbs proteins underlie an ancient program for membrane stabilization through regulation of intracellular attachment protein complexes and their content at the surface membrane, although ER expansion through ATF6α appears to have evolved phylogenetically after *Drosophila*.

## Discussion

A vast majority of the Thbs literature over the past 3 decades have invoked or interpreted data consistent with a primary extracellular function for these proteins, while only a handful have shown a direct intracellular function (*Adams and Lawler, 2011*; *Ambily et al., 2014*; *Baek et al., 2013*; *Brody et al., 2016*; *Christopherson et al., 2005*; *Duquette et al., 2014*; *Frolova et al., 2014,2010, 2012*; *Hauser et al., 1995*; *Lynch et al., 2012*; *McKeown-Longo et al., 1984*; *Posey et al., 2014*; *Schellings et al., 2009*; *Södersten et al., 2006*). Here, we identify a fundamental yet previously unrecognized intracellular role for Thbs4 in skeletal muscle, where it directly augments selective vesicular trafficking and chaperones DGC and integrin attachment complexes to the membrane, leading to greater stability and levels of select complexes at the sarcolemma, and thereby enhancing the mechanical stability of the myofiber (*Figure 9*). In fact, our findings identify Thbs4 as a crucial component to maintain muscle fiber integrity as loss of *Thbs4* results in sarcolemma weakness that causes spontaneous dystrophic changes with aging. Importantly, the Thbs protective effect holds true in both mouse and *Drosophila* skeletal muscle.

As secreted matricellular proteins, Thbs' are first produced in the ER where they are glycosylated and transit to the Golgi for additional modifications, where after they traverse the remainder of the secretory pathway (*Adams and Lawler, 2011*). Interestingly, we observed that skeletal muscle-specific overexpression of Thbs4 did not result in accumulation within the ECM, but predominantly produced an intracellular protein localization pattern within the ER and post-ER vesicular network. In fact, we have also since generated cardiac-specific Tg mice overexpressing Thbs1, 2, 3, 4 or 5 in the heart ([*Lynch et al., 2012*] and data not shown). In these hearts all five Thbs' reside mainly within the

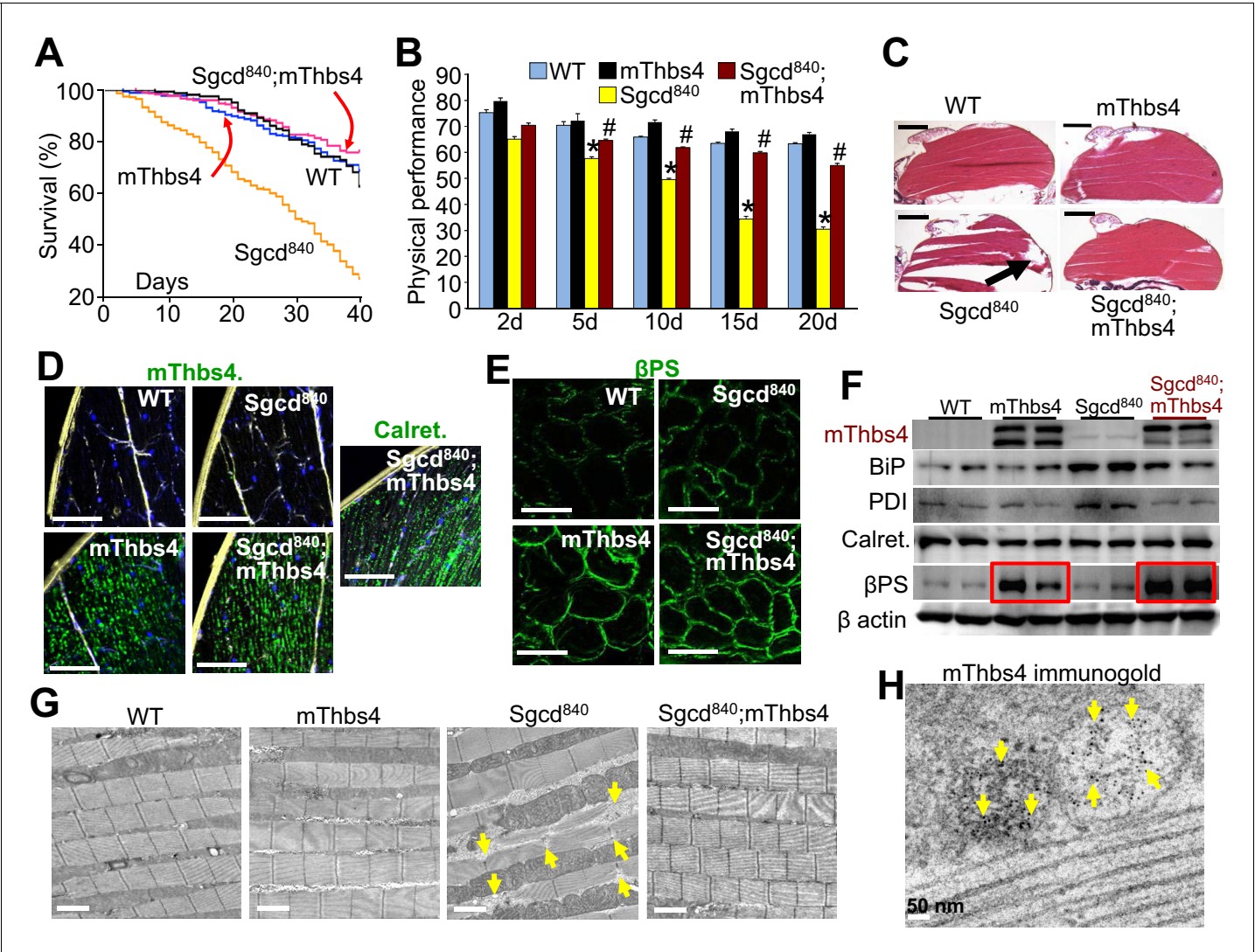

**Figure 8.** Thbs4 regulates muscle membrane integrity in *Drosophila*. (A) Survival of the *Drosophila* lines shown over a period of 40 days. p<0.001 for *Drosophila* line 840 lacking the δ-sarcoglycan homologue gene (Sgcd[840]) versus Sgcd[840];mThbs4 line that express the mouse Thbs4 protein in muscle. N = 358 for WT; 163 for mThbs4; 395 for Sgcd[840]; 300 for Sgcd[840];mThbs4. Statistical analysis performed with log rank, Mantel-Cox test. (B) Physical performance with a negative geotaxis assay of WT (n = 208), mThbs4 (n = 190), Sgcd[840] (n = 203), and Sgcd[840];mThbs4 *Drosophila* (n = 205) at the indicated ages. All data are represented as mean ± SEM. *p<0.05 vs WT and mThbs4; #p<0.05 vsSgcd[840] by one-way ANOVA with *post hoc* Tukey's test. (C) Representative H&E stained histological sections of the dorsal median indirect flight muscles at 30 days of age in the indicated genotypes of *Drosophila*. Representative images of 14 *Drosophila* per genotype studied. The arrow shows a prominent area of muscle rupture in the Sgcd[840] MD *Drosophila* line. Scale bar = 200 µm. (D) Immunohistochemistry showing increased intracellular mThbs4 and calreticulin (Calret.; both green) in longitudinal sections of the dorsal median indirect flight muscle of 30 day-old mThbs4 and Sgcd[840]; mThbs4 expressing flies compared to WT and Sgcd[840] lines. Membranes are shown in yellow, nuclei in blue. Representative images of 14 *Drosophila* per genotype studied. Scale bars = 10 µm. (E) Immunofluorescence detection of fly βPS integrin (green) in cross-sections the dorsal median indirect flight muscle of 30 day-old *Drosophila* of the indicated genotypes. Representative images of 10 *Drosophila* per genotype studied. Scale bars = 10 µm. (F) Representative Western blot of mThbs4 and ER-stress proteins BiP, PDI, calreticulin (Calret.), as well as βPS integrin in 15 day-old WT, mThbs4, Sgcd[840], and Sgcd[840];mThbs4 lines (n = 4 biological replicates). β actin was used as loading control. The red boxes show upregulation of βPS integrin with muscle specific mThbs4 overexpression in the WT and Sgcd[840] mutant lines. (G) Transmission electron microscopy of the dorsal median indirect flight muscle of WT, mThbs4, Sgcd[840], and Sgcd[840];mThbs4 *Drosophila*. Arrows indicate the characteristic sarcomeric tears and disorganization present in the Sgcd[840] line. Representative images of 4 *Drosophila* studied for the WT, mThbs4, and Sgcd[840];mThbs4 lines, and 5 *Drosophila* for the Sgcd[840] line. Scale bar = 2 µm. (H) Immunogold-transmission electron microscopy of mThbs4 in longitudinal section of the dorsal median indirect flight muscle of mThbs4 *Drosophila*. The arrows indicate mThbs4 (6 nm gold particles) localized to intracellular vesicles. Representative images of 4 *Drosophila* are shown. Scale bar = 50 nm.

*Figure 8 continued on next page*

*Figure 8 continued*

The following figure supplement is available for figure 8:

**Figure supplement 1.** Tsp expression in muscle of *Drosophila* rescues MD due to deletion of the δ-sarcoglycan-like gene (Sgcd[840]).

intracellular vesicular network and ER with only limited detectable protein accumulation outside cardiomyocytes. This is in dramatic contrast to the overexpression of an array of other matricellular proteins, such as periostin, which saturates the ECM when overexpressed (*Oka et al., 2007*). In contrast, dystrophic skeletal muscle did reveal occasional Thbs4 protein accumulation in fibrotic regions around myofibers, confirming that this protein can reside for a period of time in the ECM. These dynamic localization differences at baseline versus during fibrotic disease could be attributed to Thbs recycling at the cell surface (*Adams and Lawler, 2011*; *Wang et al., 2004*). Indeed, we observed rapid up take of recombinant Thbs4 when given exogenously to cultured C2C12 myoblasts or myotubes. It is possible that extracellular Thbs4 reuptake could occur through its designated receptor or through the integrin and DGC complexes. Indeed, other matricellular proteins such as SPARC were shown to be actively taken up back into myofibers through such a process with integrin associated endocytosis, sorting and recycling (*Chlenski et al., 2011*; *De Franceschi et al., 2015*; *Nakamura et al., 2014*).

Our data identify ATF6α as a transcriptional regulator of secretory pathway activity in muscle, a function that was previously established for the inositol-requiring enzyme 1α / X-box binding protein (IRE1α/XBP-1) axis of the canonical ER stress response pathway during plasma cell differentiation (*Shaffer et al., 2004*). Importantly, although enhancement of the ATF6α-mediated adaptive ER stress pathway was sufficient to drive the dramatic expansion of the ER and post-ER vesicular content and increase vesicular trafficking to the membrane, it did not augment membrane residency of DGC and integrin attachment complexes at the sarcolemma, nor was it sufficient to protect against MD. Thus, ATF6α is only part of a more complex intracellular mechanism whereby Thbs4 regulates mechanical stability of the muscle fiber and its sarcolemma, in coordination with greater vesicle formation and trafficking (*Figure 9*).

Data from both *Drosophila* and zebrafish show that thrombospondin proteins localize outside of cells within the tendinous junctions (*Chanana et al., 2007*; *Subramanian and Schilling, 2014*; *Subramanian et al., 2007*). In addition, vertebrate *Thbs1* and *Thbs2* genes have evolved domains that are tailored to affecting processes outside the cell, such as altering transforming growth factor-β activity and the angiogenic response (*Adams and Lawler, 2011*; *Bornstein, 2001*; *Carlson et al., 2008*). Hence, the simplest interpretation of our data and that reported in the literature is that Thbs proteins are complex, multifactorial proteins that function both inside and outside the cell. However, our working hypothesis is that Thbs4 appears more tailored to intracellular functionality in cardiac and skeletal muscle (*Figure 9*), and this same paradigm appears to hold true for the other Thbs family members in other tissues. For example, Thbs1 was previously shown to localize to the intracellular side of membrane attachment sites by immunogold-electron microscopy in endothelial cells (*Hiscott et al., 1997*). Furthermore, Thbs proteins are known to strongly interact with many different integrin heterodimers, and singular proteomic analysis of the α5β1 integrin complex identified Thbs2 as a core element of its 'interactome' (*Adams and Lawler, 2011*; *Bouvard et al., 2013*; *De Franceschi et al., 2015*; *Plow et al., 2000*; *Schiller et al., 2013*). Our ultrastructural and biochemical analyses showed that the Thbs-integrin complex resides within the lumen of vesicles. Although further, in-depth analyses will suggest whether Thbs4 co-regulates the signaling competence of this integrin complex and the exact preassembly stage that is influenced by Thbs4 on the way to the cell surface. Finally, Thbs1 silencing or overexpression in human cancer cells was shown to decrease or enhance integrin protein levels, respectively, in the intracellular and plasma membrane compartment and thereby modulate cellular adhesion (*Duquette et al., 2013*; *John et al., 2010*).

Taken together, various lines of evidence indicate that Thbs proteins stabilize integrins and the DGC at membrane attachment complexes of the sarcolemma through an intracellular function. Importantly, this observation holds true in both mouse and *Drosophila* skeletal muscle where integrin protein content at the cell membrane was increased with Thbs4 overexpression, yet very little Thbs

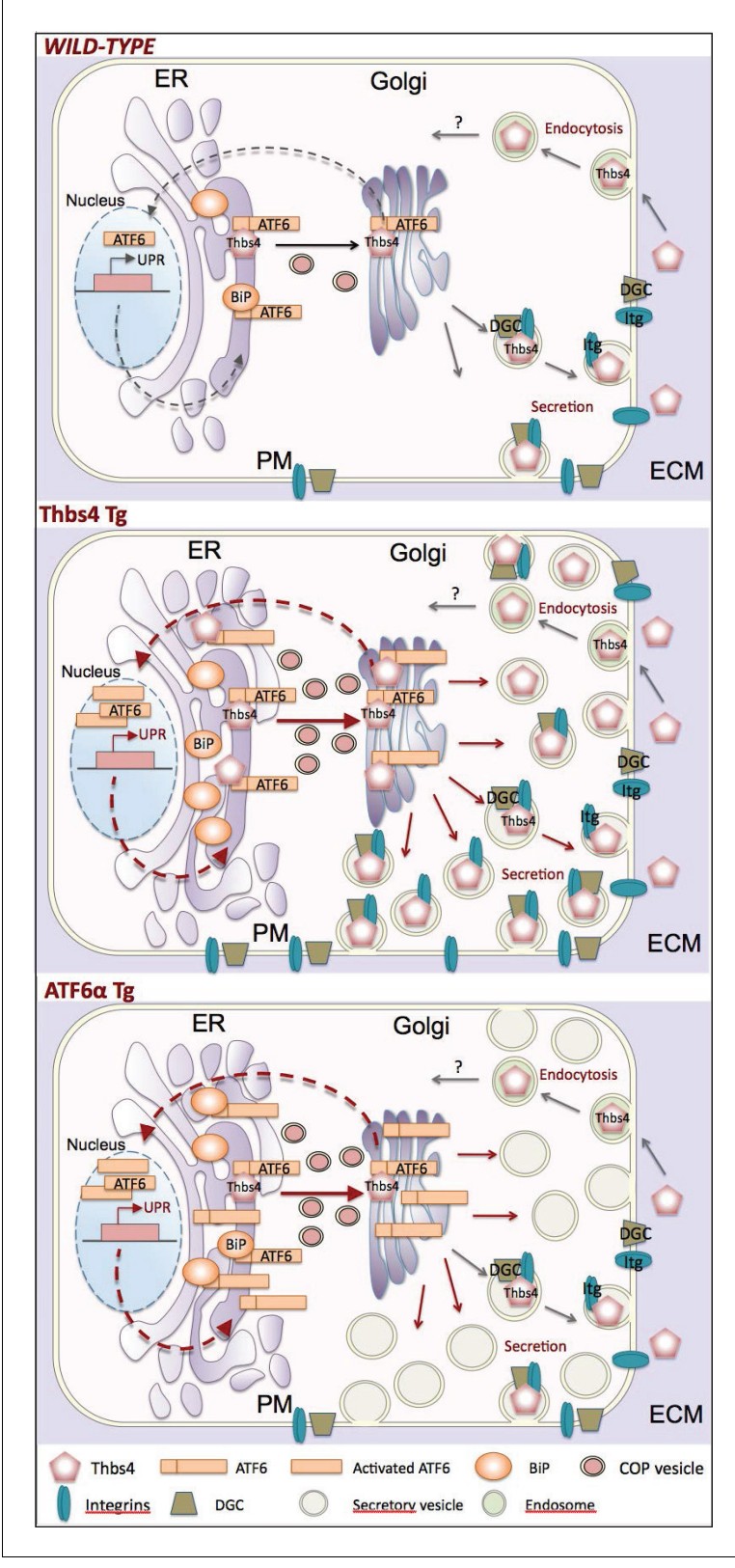

**Figure 9.** Model of how Thbs4 functions as an intracellular regulator of muscle cellular attachment and membrane stability. As a matricellular protein, thrombospondin-4 (Thbs4) pentamers are synthesized in the ER lumen and then transported to the Golgi, where after they traverse the secretory pathway to fuse with the plasma membrane for secretion. Thbs4 can then reside within the extracellular matrix (ECM) or be actively endocytosed and returned

*Figure 9 continued on next page*

*Figure 9 continued*

to the intracellular compartment by a recycling receptor. In addition to its established extracellular functions, combined studies in the heart and skeletal muscle now reveal that while in the ER, Thbs4 can compete with BiP (GRP78) for binding to the ER-resident transcription factor ATF6α, thereby facilitating ATF6α translocation to the Golgi for processing and subsequent shuttling to the nucleus where it regulates expression of ER stress responsive genes that are also part of the unfolded protein response (UPR). ATF6α induction in cardiac and skeletal muscle, or by overexpression in Tg mice (lower panel) causes a dramatic expansion of the ER and post-ER vesicles, as well as increased vesicular trafficking to the membrane. The ability of Thbs4 to induce ATF6α processing and nuclear trafficking also causes this same ER expansion and augmentation of intracellular vesicular trafficking to the membrane. However, Thbs4 uniquely regulates trafficking of selected integrins and dystrophin-associated glycoprotein complexes (DGC) members to the sarcolemma, thereby enhancing the mechanical stability of the myofiber, such as observed in Thbs4 Tg muscle (middle panel).

protein was observed outside the cell in either species at baseline. While previously published data showed a large concentration of *Drosophila* Tsp protein to the myotendinous junction in stage 16 embryos, this is also the very same region where the integrins are highly concentrated in identical foci and the data do not distinguish if Tsp in inside or outside the cell (*Subramanian et al., 2007*). At earlier embryonic stages (stage 12–13) however, the Tsp protein is intracellular with a diffuse pattern similar to that of integrins that have yet to be deposited at the cell membrane (*Subramanian et al., 2007*). Moreover, in later stage *Drosophila* larvae Tsp is no longer detected in the myotendinous junctions or in tendons. Rather, a network of Tsp positive staining is observed within the muscle of larvae (*unpublished observations*, Talila Volk). However, in mammals Thbs4 can oligomerize with Thbs5 and be deposited in the tendon (*Hauser et al., 1995*; *Södersten et al., 2006*), loss of *Thbs4*$^{-/-}$ in mice showed altered ECM composition and weakened muscle-tendons (*Frolova et al., 2014*), and altered inflammatory responses associated with arteriogenesis (*Frolova et al., 2010*), whereas Thbs1 and Thbs2 were shown to function from outside the cell in augmenting developmental synaptogenesis (*Christopherson et al., 2005*). Hence, Thbs proteins are clearly complex regulatory proteins with intra- and extracellular functions.

One aspect of the biology that was not conserved in *Drosophila* was the ability of Thbs or Tsp overexpression to expand the intracellular vesicular compartment in this lower organism, likely because *Drosophila* does not rely on an ATF6-like mechanism for the ER stress response, and the Thbs interacting domain within ATF6α is not contained in the *Drosophila* homologue of this gene (*Mori, 2009*). Thus, the ability of Thbs proteins to activate ATF6α to augment ER protein production and secretory pathway activity is a later evolutionary adaptation beyond just stabilizing membrane attachment complexes, which in higher organisms more effectively coordinates tissue remodeling and healing through the Thbs proteins.

Taken together, the unique aspects of muscle membrane biology allowed us to uncover a previously unknown and possibly dominant intracellular function of the Thbs proteins that is evolutionarily conserved (*Figure 9*). This new model for Thbs protein function has many disease ramifications, especially in skeletal muscle, a tissue that is highly sensitive to mutations in genes that cause weaknesses in cellular attachment and membrane stability. Indeed, there is a lack of therapeutic strategies to effectively treat MD and our study suggests that this protein may provide a universal approach to strengthen the sarcolemma in skeletal muscle if employed in a gene therapy approach. Importantly, this protein is already present in muscle and it would not be perceived as a neo-antigen by viral-mediated overexpression, hence it should be well tolerated and best used in patients where the genetic basis of their MD disease is due to a loss of structural support of the sarcolemma (majority of cases).

## Materials and methods

### Mouse models

Skeletal muscle-specific transgenic mice for Thbs4 and ATF6α were generated using the modified human skeletal α-actin (Ska) promoter construct as previously described (*Goonasekera et al., 2011*; *Lynch et al., 2012*). Briefly, full-length mouse Thbs4 cDNA was obtained from Open Biosystems

(Accession number: BC139414) and amplified by PCR and cloned into the *BamHI* and *EcoRV* sites of the Ska-promotor expressing vector (forward: 5'-CGCGGATCCATGCCGGCCCCACGCGCG-3', and reverse: 5'-ATCTCAATTATCCAAGCGGTC AAAACTCTGGG-3'). Full-length mouse ATF6α cDNA was obtained from a previously generated pcDNA1-ATF6α plasmid (*Lynch et al., 2012*). Mouse ATF6α cDNA was amplified by PCR and subsequently cloned by PCR into the *KpnI* and *NotI* sites of the Ska-promotor expressing vector (forward: 5'- GGGGTACCATGGAGTCGCCTTTTAGTCC-3', and reverse: 5'-ATAAGAATGCGGCCGCCTACTGCAACGACTCAGGGAT-3'). All constructs were confirmed by DNA sequencing.To make Tg mice, the Ska-plasmid backbone was removed and the Ska-Thbs4 and Ska-ATF6α fragments were gel purified followed by Elutip-D column purification (Schleicher and Schuell Bioscience; Dassel, Germany, Cat. 10462617) for newly fertilized oocyte injection at the Cincinnati Children's Hospital Transgenic Animal and Genome Editing Core Facility. All transgenic mice were produced in the FVB/N background. Mice deficient for Thbs4 (*Thbs4*[-/-]; Strain: B6.129P2-Thbs4tm1Dgen/J) and *mdx* mice (Strain: C57BL/10ScSn-*Dmd*[mdx]/J) were purchased from Jackson Laboratories (Bar Harbor, Maine). *Sgcd*[-/-] mice were previously described (*Hack et al., 2000*). Next, Ska-Thbs4-Tg, ska-ATF6α-Tg and *Thbs4*[-/-] mice were backcrossed for at least 6 generations into the *Sgcd*[-/-] background to generate *Sgcd*[-/-] Thbs4-Tg mice, *Sgcd*[-/-] ATF6α-Tg mice and *Sgcd*[-/-] *Thbs4*[-/-] mice, as well as their littermate controls. An identical breeding strategy was used to generate *mdx Thbs4*[-/-] mice. In addition, males from each transgenic line were crossed to *mdx* heterozygous females to generate *mdx*-Tg and *mdx* non-Tg male littermates and their appropriate controls. All animal experiments were approved by the Institutional Animal Care and Use Committee of the Cincinnati Children's Hospital Medical Center (Protocol# IACUC2013-0013). No human subjects or human tissue was directly used in experiments in this study.

## Thbs4 mRNA expression levels in various human muscle diseases

A search of the 'National Center for Biotechnology Information Gene Expression Omnibus (NCBI GEO)' database (*Barrett and Edgar, 2006*) revealed that Bakay and colleagues recently performed microarray experiments on human muscle biopsies of various muscle diseases (GEO accession GDS1956/204776, [*Bakay et al., 2006*]). Available data included 11 different muscle diseases, with a total of 121 human muscle biopsy specimens tested on Affymetrix U133A microarrays. Individual Thbs4 mRNA levels from samples with Becker Muscular Dystrophy (BMD, n = 5); Duchenne Muscular Dystrophy (DMD, n = 10); dystrophy due to calpain-3 mutations (LGMD2A, n = 10); and dystrophy due to a paucity of dysferlin (LGMD2B, n = 10) were averaged and compared to those from healthy muscle biopsies (n = 18) using an unpaired two-tailed *t*-test.

## Adeno-associated virus (AAV) serotype-9 production and injection (gene-therapy)

Mouse Thbs4 or an eGFP cDNA was amplified by PCR and inserted into the *BamHI* and *XhoI* sites of pAAV-MCS vector. AAV9-CMV-eGFP and AAV9-CMV-Thbs4 were produced using the triple transfection method in HEK293 cells as previously described and stored at −80°C until commencing the in vivo experiments (*Gray et al., 2011*; *Zincarelli et al., 2008*). Next, both left and right gastrocnemius muscles of three-day-old *Sgcd*[-/-] mice were injected with either AAV9-Thbs4 or AAV9-eGFP (both 1E10 viral particles in 30 µl isotonic saline; [*Goonasekera et al., 2011*]). Mice were sacrificed at six weeks of age. The left gastrocnemius of each mouse was fixed, processed, paraffin embedded, sectioned, and stained with H&E and Masson's trichrome, whereas the right muscles were snap-frozen in liquid nitrogen for storage in −80°C. A subset of muscles were embedded in Optimal Cutting Temperature Compound (O.C.T, Tissue-Tek, Sakura Americas, Torrance, CA, Cat #4583), frozen, and 7 µm cryosections were generated to confirm eGFP expression by direct fluorescence (not shown).

## Histological analysis and immunohistochemistry

Indicated muscles were fixed overnight in 4% paraformaldehyde, dehydrated in ethanol and paraffin embedded. Global muscle architecture and pathological indices were determined from 5 µm thick transverse sections at the center of the muscle stained with either H&E or Masson's trichrome (*Goonasekera et al., 2011*). Approximately 1000 fibers per mouse for each muscle group were counted for analysis of percentage central nucleation using ImageJ software. Interstitial fibrosis was

quantified using ImageJ software as percentage of blue area in Masson's trichrome stained paraffin-embedded sections (*Goonasekera et al., 2011*). For co-labeling of tissue sections with Thbs4 and ER marker calreticulin on paraffin-embedded muscles, 5 µm sections were rehydrated and heated in 1x antigen retrieval CITRA (BioGenex, Fremont, CA, Cat# HK086-9K). Muscle sections were permeabilized for 10 min in 0.3% triton/PBS and then in a blocking buffer (0.1% triton/PBS, 5% goat serum, 2% BSA) for 1 hr at room temperature. Primary antibody incubations were overnight at 4°C (Thbs4: AF2390, 1:150 dilution, R&D Systems, Minneapolis, MN; and ER-marker calreticulin: Abcam, ab2907, Cambridge, MA, 1:100; all in blocking buffer). Appropriate Alexa Fluor-488 (green) and Alexa Fluor-568 (red) secondary antibodies (Invitrogen, Waltham MA, 1:400 in blocking buffer) were applied for 2 hr at room temperature and subsequently for 10 min with DAPI nuclear DNA stain (Invitrogen, 1:10.000). For co-labeling of tissue sections with Thbs4 and collagen I or periostin, freshly harvested quadriceps were fixed for 4 hr in 4% PFA at 4°C, rinsed with PBS and cryoprotected in 30% sucrose/PBS overnight before embedding in O.C.T. Afterwards, 10 µm cryosections were collected, rinsed in PBS and blocked for 30 min at room temperature in blocking solution (PBS with 5% goat serum, 2% bovine serum albumin, 0.1% Triton X-100). Primary antibody incubations were overnight at 4°C (Thbs4: AF2390, 1:150 dilution, R&D Systems, Minneapolis, MN; collagen type I, 1:300 dilution, Abcam, ab34710; periostin, NBP1-30042, Novus Biologicals, Littleton, CO; 1:200 dilution; all in blocking buffer). Next, Alexa Fluor-488 (green) and Alexa Fluor-568 (red) secondary antibodies (Invitrogen, Waltham MA, 1:400 in blocking buffer) were applied for 2 hr at room temperature and subsequently for 10 min with DAPI nuclear DNA stain (Invitrogen, 1:10.000).

For IHC of the dystrophin-glycoprotein complex (DGC)-associated proteins, freshly harvested quadriceps from three month-old mice were embedded in O.C.T and frozen in liquid nitrogen. Transverse tissue sections were cut at a thickness of 7 µm and stored in −80°C until further use. Here sections were acclimated to RT for 15 min, post-fixed in ice-cold methanol for 10 min, washed in PBS and then blocked for 30 min with either 0.1% triton/PBS, 5% goat serum, 2% BSA or with Mouse on Mouse (M.O.M.) blocking reagent (Vector Laboratories, Burlingame, CA, BMK-2202) as described by the manufacturer's protocol for mouse primary antibodies. Sections were incubated with primary antibody in blocking solution at 4°C overnight. Primary antibodies included: δ-sarcoglycan (Abcam, ab92896, 1:100), α-sarcoglycan, β-sarcoglycan, γ-sarcoglycan (NovaCastra, Buffalo Grove, IL, NCL-a-sarc, NCL-b-sarc and NCL-g-sarc, all 1:250), β-dystroglycan (Development Studies Hybridoma Bank, Iowa City IA, MANDAG2; 1:50), utrophin (Santa Cruz Biotechnology, MANCHO7, sc-81557; 1:50), dystrophin (Abcam, ab15277, 1:200) and β1D-integrin (Millipore, Billerica, MA, MAB1900, 1:250). Primary antibodies were detected by applying Alexa Fluor-488 conjugated goat-anti-rabbit (Invitrogen, 1:400 in blocking buffer) for 2 hr at RT or biotinylated anti–mouse (Vector Laboratories, M.O.M kit, 1:500) followed by Alexa Fluor-488 streptavidin conjugate (Invitrogen, 1:200), both for 45 min at RT. For each immunostain, sections were incubated with secondary and tertiary antibodies alone as a control for specificity (not shown). All sections were mounted in Vectashield Hard Set (Vector Laboratories, H-1400) to prevent photobleaching and visualized using a Nikon A1 confocal laser microscope system equipped with 40x H$_2$O objective (NA = 1.15). All imaging was done under identical conditions using NIS Elements Advanced Research (AR) microscope imaging software (Nikon Instruments Inc. Melville, NY).

## Protein preparations and western blotting

Quadriceps, gastrocnemius, soleus, diaphragm and hearts were harvested and immediately frozen in liquid nitrogen for storage at −80°C. To evaluate ER-stress and Thbs4 protein expression, muscles were homogenized (Fisher Scientific, Waltham, MA, TissueMiser) in ice-cold RIPA buffer containing Halt Protease Inhibitor cocktail (ThermoScientific, Waltham, MA, #78430). Next, samples were sonicated (SP Scientific, Warminster, PA, VirSonic 60, power setting 3 for 3 times 10 s), lysates were cleared by centrifugation at 14,000 rpm for 14 min at 4°C and stored at −80°C.

To evaluate glycosylation pattern, quadriceps protein extracts were treated with Endoglycosidase H (Endo H; New England Biolabs Inc., Ipswich, MA, P07P2), peptide N-glycosidase F (PNGase F, New England Biolabs Inc., P0704) or protein deglycosylation mix (New England Biolabs Inc., P6039) prior to SDS-PAGE, according to the manufacturer's instructions. Endo H cleaves high mannose residues at hybrid oligosaccharides present on proteins in the ER, whereas PNGase cleaves both these and more complex oligosaccharides that result from processing in the Golgi (*Hewett et al., 2004*). Control samples were treated the same way without addition of enzymes.

To evaluate DGC-associated proteins, fresh quadriceps muscle was harvested and crude sarco-lemmal isolates were prepared as previously described (*Kobayashi et al., 2008*). Briefly, freshly harvested quadriceps was homogenized in 7.5x volumes of ice-cold lysis buffer (20 mM $Na_4P_2O_7$, 20 mM $NaH_2PO_4$, 1 mM $MgCl_2$, 0.303 M sucrose, 0.5 mM EDTA, pH 7.1 with 5 µg/ml aprotinin and leupeptin, 0.5 µg/ml pepstatin A, 0.23 mM PMSF, 0.64 mM benzamidine, and 2 µM calpain inhibitor I and calpeptin), then centrifuged 14,000 g for 20 min at 4°C; the pellet was re-suspended, re-homogenized and both supernatants were centrifuged 30,000 g for 30 min at 4°C after which the pellet was re-suspended in 100 µl lysis buffer and stored at −80°C until further use.

Extracellular protein fractionation from quadriceps muscle was essentially performed as previously described (*Tjondrokoesoemo et al., 2016*). In summary, freshly harvested quadriceps muscle was minced, washed with PBS and subjected a 1 hr washing step in 0.5 M NaCl, 10 mM Tris-HCl, 25 mM EDTA (PH 7.5). Next, samples were decellularized overnight in 0.1% SDS, 25 mM EDTA, followed by extracellular matrix extraction with 4 M guanidine hydrochloride, 50 mM $C_2H_3NaO_2$, 25 mM EDTA (pH 5.8). Finally, proteins were precipitated overnight in 80% EtOH, air dried and treated with protein deglycoylation mix (New England Biolabs Inc., P6039).

Immunoprecipitations were performed as described (*Brody et al., 2016*). Briefly, rat neonatal cardiomyocytes were transduced with adenovirus to overexpress Thbs4 with a C-terminal Flag tag or a β-galactosidase (βgal) expressing construct as a control. Two days later, cardiomyocyte lysates were harvested and immunoprecipitated with anti-Flag magnetic beads (Sigma-Aldrich, M8823). Immuno-precipitates were resolved by SDS-PAGE, transferred to PVDF membranes, and immunoblotted for Flag (Cell Signaling Technology, 2368; 1:1000); β1D-integrin (Millipore, MAB1900, 1:500); α7-Integrin (Santa Cruz Biotechnology, sc-27706; 1:200); and β-dystroglycan (Development Studies Hybridoma Bank, MANDAG2 clone 7D11; 1:100).

β1D-integrin positive intracellular vesicles were isolated from quadriceps using an endoplasmic reticulum isolation kit (Sigma Aldrich, ER0100), according to the manufacturer's instructions. Briefly, tissues were homogenized in isotonic extraction buffer using a 2 ml Dounce homogenizer. Homogenates were cleared by centrifugation at 12,000 g for 15 min at 4°C. Two mg of vesicles was incubated with for 12 hr at 4°C with an antibody raised against the cytoplasmic domain of β1D-integrin (Millipore, MAB1900), immunoprecipitated using A/G magnetic beats (ThermoFisher Scientific, #88803) at 4°C for 1hr, and subsequently subjected to SDS-PAGE.

All protein concentrations were determined using DC Protein Assay Kit (Bio-Rad, Hercules, CA, #5000111). Then, 5X Laemmli buffer was added to protein preparations, which were then heated to 95°C for 5 min, and equal quantities were subjected to SDS-PAGE. In all instances, the wet transfer method was utilized with PVDF membranes (Millipore, IPVH00010). Staining of non-specific bands on PVDF membranes with Ponceau S solution (Sigma-Aldrich, P7170) was used as a loading control for sarcolemmal isolates. Approximately 5% blotto (nonfat dry milk [Carnation] in TBS with 0.2% Tween 20 [ThermoFisher Scientific]) was used to block membranes for 45 min at RT and to incubate the membranes in primary antibodies overnight at 4°C. Two primary antibodies used to detect Thbs4 (R&D Systems, AF2390 shown in *Figure 1D*, or Santa Cruz Biotechnology, sc-7657-R shown in all other figure panels as a doublet for Thbs4; 1:1000 dilution for both) in our transgenic and gene deleted muscles. For ATF6α (Abcam, ab37149 at 1:1000), the 50 kDa cleaved active form, which resides primarily in the nucleus, was shown in all the western blots. Other primary antibodies used in this study included: α-actinin (sarcomeric, A7811; 1/1000); Armet (Abcam, ab67271; 1:1000); BiP (Cell Signaling Technology, Danvers, MA, 3177; 1:1000); calreticulin (Cell Signaling Technology, 2891; 1:1000); dihydropyridine receptor α1 (Cav1.1; Thermo Fisher Scientific, MA3-920; 1:1000); dysferlin (Abcam, ab124684 [JAI-1-49-3]; 1:1000); CLIC3(Santa Cruz, sc-390006; 1:200); α-dystroglycan (EMD Millipore, 05–593; 1:500); β-dystroglycan (Development Studies Hybridoma Bank, MANDAG2 clone 7D11; 1:100); dystrophin (Sigma-Aldrich, D8043; 1:1000); gadph (Fitzgerald, Acton, MA, 10R-G109A; 1:10000); GFP (Abcam, Ab290; 1:1000); Flag (Cell Signaling Technology, 2368; 1:1000); α5-integrin (EMD Millipore, Billerica, MA, AB1928; 1/1000); α7-integrin (Abacm, ab203254; 1:1000); β1D-integrin (Millipore, MAB1900, 1:500); Laminin (Abcam, ab11575; 1/1000); Pdi (Cell Signaling Technology, 2446; 1:1000); rab3 (Abcam, ab3336; 1:1000); rab4 (Cell Signaling Technology, 2167; 1:1000); rab5 (Sigma-Aldrich, R7904; 1:500); rab6 (Cell Signaling Technology, 4879; 1:1000); rab7 (Cell Signaling Technology, 9367; 1:1000); rab8 (Abcam, ab188574; 1:1000); rab11 (Abcam, ab95375; 1:1000); rab24 (BD Biosciences, Franklin Lakes, NJ, 612174; 1:1000); Sar1 (Abcam, ab125871; 1:1000); α-sarcoglycan (Development Studies Hybridoma Bank, IVD3(1)A9; 1:100); β-

sarcoglycan (Novus Biologicals, Littleton, CO, NBP1-90300; 1:1000); δ-sarcoglycan (Abcam, ab137101; 1:500); sarcospan (Santa Cruz Biotechnology, sc-393187; 1:200); α-tubulin (Santa Cruz Biotechnology, sc-8035; 1:1000 dilution); utrophin (Development Studies Hybridoma Bank, MAN-CHO3(8A4); 1:50) and Wash1 (Abcam, ab157592; 1:1000). In all instances, appropriate IgG-AP conjugated secondary antibodies (Santa Cruz Biotechnology, 1:2500) were utilized and membranes were either exposed to ECF substrate (Amersham Biosciences, GE Healthcare, Buckinghamshire, England, RPN5785) and visualized using a Gel-Doc XR+ system with Image Lab Software (Bio-Rad Laboratories) or probed with the appropriate secondary antibodies and visualized using the Odyssey CLx Imaging System (both Li-COR Biosciences, Lincoln, NE). Semi-quantitative analysis presented in *Figure 1—figure supplement 1* and *Figure 7—figure supplement 1* was performed using ImageJ software.

## Forced treadmill running, Evan's Blue Dye (EBD) uptake and serum CK levels

To assess the exercise capacity of mice and sarcolemmal stability, mice were subjected to forced treadmill running in the presence of EBD as previously described (*Goonasekera et al., 2011*). Briefly, adult mice were intraperitoneally injected with EBD (10 mg/ml; 0.1 ml per 10g body weight) and 24 hr later subjected to forced downhill treadmill running to measure membrane rupture events. For the exercise protocol, exhaustion of the mice was assessed as greater than 10 consecutive seconds on the shock grid without attempting to re-engage running on the treadmill. Mice were then sacrificed and quadriceps, gastrocnemius and diaphragm was embedded in O.C.T. and frozen in liquid nitrogen. Tissue sections were cut at a thickness of 7 μm, air-dried, washed in PBS and stained with wheat germ agglutinin conjugated to FITC (Sigma-Aldrich, green) for 1 hr at RT to visualize the membranes. Images were taken on a Nikon Eclipse Ti-S inverted microscope system equipped with NIS Elements Advanced Research (AR) microscope imaging software (Nikon Instruments Inc. Melville, NY) to determine the percentage of EBD-positive fibers. In addition, blood was taken from a separate cohort of un-exercised mice of each genotype to evaluate their baseline serum CK levels as previously described at the clinical laboratory of Cincinnati Children's Hospital Medical Center by an observer blinded to the genotypes (*Kobayashi et al., 2008*).

## Lengthening contraction based injury and isometric muscle force measurements

Mice were anesthetized with an intraperitoneal injection of pentabarbitol and placed supine on the muscle testing apparatus (Aurora Scientific, Aurora, ON, Canada). A midline incision running from the ankle to the thigh was created and the skin and fascia was gently removed leaving the tibialis anterior (TA) muscle exposed. The leg was immobilized by securing it in a custom jig (Aurora Scientific) with thumbscrews at the distal femur. A 4–0 nylon suture was tied to the distal TA securing it with a small plastic ring at the muscle tendon junction. The distal tendon was transected and the TA elevated to remove its contact with the tibia, and the muscle was mounted to a servomotor (Aurora Scientific, 305C) using the plastic ring. Two intramuscular electrodes were placed on either side of the peroneal nerve and stimulation voltages and optimal muscle length ($L_0$) were determined and then adjusted to produce maximal isometric force ($P_0$) at a stimulation frequency of 200 Hz. Five consecutive isometric contractions were averaged as a measure of the maximal specific tension. An additional lengthening contraction injury protocol was added in which an isometric contraction was performed to determine baseline force generation, followed by 2 consecutive 20% $L_0$ lengthening contractions, and finally another isometric contraction performed at $L_0$. The force deficit was calculated between the first isometric contraction and those following the lengthening injury cycles (See *Figure 4A* for schematic representation). This contraction injury protocol was repeated 2 more times and the force deficit was calculated relative to the pre-injury isometric contraction. In a subset of experiments passive tension was measured prior to the lengthening contraction protocol. Here the TA was set to $L_0$ and passively stretched to 5, 10, 15, and 20% of $L_0$. Each stretch was held for 2 min before the length was returned to its starting position. Maximal passive tension was recorded at the peak of the stretch. A 2-min rest period occurred between each contraction for all experiments and force values were normalized to the muscle's physiologic cross-sectional area. In some cases we

observed tendon breaks during the lengthening contraction protocol. Percentage of tendon breaks, assessed by complete physical rupture of the tendon at the muscle aponeurosis, was recorded.

## C2C12 recombinant Thbs4 labeling and In vitro Thbs4 internalization assays

C2C12 mouse myoblasts (ATCC, ATCC-1772) were plated in either Ibidi μ-slide 8-well dishes (Ibidi USA, Cat# 80826) or in 6-well plates (Fisher Scientific, Cat# 353046) and maintained in DMEM/high glucose (Fisher Scientific, Cat# SH30022.01) supplemented with 10% bovine growth serum (Fisher Scientific, Cat# SH3054103) and 1x penicillin-streptomycin (Cellgro 30-0002-CI, Mediatech, Corning Life Sciences) at 37°C in 5% $CO_2$. Cells were kept to a maximum of eight passages. For differentiation, cells were grown to confluence and then switched to DMEM/high glucose supplemented with 2% donor equine serum (Life Technologies, Cat# 26050088) and 1x penicillin-streptomycin for five days.

Alexa Fluor-488 Microscale Protein Labeling Kit (Life Technologies, Cat#30006) and EZ-Link Sulfo-NHS-Biotin (Thermo Fisher Scientific, Cat# 21217) were used to label recombinant mouse Thbs4 (rThbs4, R&D Systems, Cat# 7860-TH-050) according to the manufacturer's instructions. Equal amounts of Alexa-488 labeled bovine serum albumin (BSA, Fisher Scientific, Cat# BP1605-100G) were used as control.

To assess internalization of rThbs4 by cultured C2C12 myoblasts and myotubes, established approaches were used as previously described (*Chlenski et al., 2011*; *Nakamura et al., 2014*). First, cells plated in Ibidi μ-slide 8-well dishes were treated with either 1 μg/ml Alexa-488 labeled rThbs4 or equal amounts of Alexa-488 labeled BSA control for the indicated periods. Next, cells were rinsed with sterile PBS and fixed with 4% paraformaldehyde for 10 min. After fixation cells were washed three times with PBS, followed by blocking with 3% normal goat serum/PBS/0.1% Triton for 20 min at room temperature and subsequently incubated with Alexa Fluor-568 labeled phalloidin (Life Sciences, Cat# A12380; 1/100 in blocking buffer) at room temperature to visualize the F-actin cytoskeleton or incubated with anti-Rab7 (late endosomes; Cell Signaling, #9367; 1/100 in blocking buffer) overnight at 4°C, followed by an Alexa fluor-568 (red) secondary antibody for 45 min (Invitrogen, 1:400 in blocking buffer). In both conditions nuclei were counterstained with DAPI nuclear DNA stain (Invitrogen, 1:10.000), mounted in Ibidi mounting medium for fluorescent microscopy (Ibidi USA, Cat# 50001) and visualized using a Nikon A1 confocal laser microscope system (Nikon Instruments Inc. Melville, NY) as described above. In parallel, cells plated in 6-well plates were treated with either 1 μg/ml Biotin labeled rThbs4. Next, cells were rinsed with PBS, lysates were prepared as described above and intracellular biotin labeled proteins were visualized and quantified by western blot analysis using Streptavidin DyLight 650 conjugate (ThermoFisher Scientific, Cat# 84547) at a 1:1000 dilution on the Odyssey CLx Imaging System (Li-COR Biosciences, Lincoln, NE). All experiments described above were performed in triplicate.

## Muscle fiber isolation and laser induced membrane injury

Flexor digitorum brevis (FDB) muscle fibers were isolated from male age-matched mice of each genotype as previously described and plated onto 35 mm glass-bottomed MatTek dishes (MatTek Corp., Ashland MA, P35G-0-10-C) in isotonic Tyrode buffer containing 1.25 mM $Ca^{2+}$ (*Cai et al., 2009*). Membrane damage was induced in the presence of 2.5 μM FM1-43 dye (Molecular Probes, Eugene OR) using a Nikon A1 confocal laser microscope through a Plan Apo 60x $H_2O$ immersion objective. To induce damage, a 5x5 pixel area of the sarcolemma on the surface of the muscle fiber was irradiated using a UV laser at full power (80 mW, 351/364) for 10 s at t = 60 s (*Cai et al., 2009*). Images were captured 5 min after irradiation at 5-second intervals (*Cai et al., 2009*). For each image, fluorescence intensity in an area of about 200 μm$^2$ directly adjacent to the injury site was measured using ImageJ software. To allow for statistical analysis from different experiments, data are presented as fluorescence intensity relative to the value before injury (ΔF/F0).

## Adenoviruses

Recombinant adenoviruses harboring Thbs4, the N-terminal Laminin G (Ad-LamG) domain of Thbs4, the Type III repeat (Ad-T3R) domain of Thbs4, a Thbs4 $Ca^{2+}$-binding mutant containing mutations in six DXDXDG calcium-binding sites within the T3R domain of mouse Thbs4 (Ad-Thbs4-mCa$^{2+}$),

constitutively-nuclear ATF6α (Ad-ATF6α-CN, amino acids 1–364), the ER luminal domain of ATF6α (amino acids 448–570) with a C-terminal KDEL ER retention signal (Ad-ATF6α-DN) and β-gal control were previously generated and validated (*Brody et al., 2016*; *Lynch et al., 2012*). Adenoviral LamG, T3R and Thbs4-mCa$^{2+}$ were engineered to contain the N-terminal signal peptide and the coiled-coil domain of Thbs4 to ensure proper intracellular trafficking and assembly and oligomerization in the ER (*Brody et al., 2016*). The cDNAs of human Nell2 (Harvard Plasmids, HsCD00331038) and eGFP-tagged VSV-G-ts045 (VSVG-eGFP, Addgene: Plasmid #11912 deposited by Dr. Jennifer Lippincott-Schwartz) were amplified by PCR for insertion into the pAdenoX-CMV vector (Clontech, Mountain View, CA) and transfected into HEK cells to generate recombinant adenovirus following manufacturer's instructions (*Brody et al., 2016*; *Patterson et al., 2008*).

## In vivo adenoviral transduction and tissue processing

Experiments were performed as previously described (*Brody et al., 2016*). Briefly, either purified AdThbs4-Flag, AdThbs4-mCa$^{2+}$-Flag or Adβgal control were injected into the left and right gastrocnemius muscle of individual one-day-old Sprague Dawley rat pups (Envigo, Indianapolis IN, USA), followed by an additional injection 48 hr later ($10^8$ viral particles for each injection). Rat pups were sacrificed at eight days of age and muscles were embedded in O.C.T. frozen, and 10 μm cryosections were generated. To visualize adenoviral transduction (Flag) and the effects of our constructs on the membrane residency of β1 integrin, tissue sections washed three times with PBS, followed by blocking with 3% normal goat serum/PBS/0.1% Triton for 30 min at room temperature and then incubated with anti-flag and anti-β1 integrin primary antibodies (Cell Signaling, #2368; 1:500 and EMD Millipore, MAB1997, 1:100, respectively, in blocking buffer) overnight at 4°C. Primary antibodies were detected by applying Alexa Fluor-488 conjugated goat-anti-rabbit and biotinylated anti–mouse (Vector Laboratories, M.O.M kit, 1:500) followed by Alexa Fluor-568 streptavidin conjugate (Invitrogen, 1:200) for 45 min at RT. Sections were mounted in Vectashield Hard Set (Vector Laboratories, H-1400) to prevent photobleaching and imaged as described above. Sarcolemmal localized β1 integrin was quantified as percentage of red area per square surface of adenoviral transduced (Flag-positive) myofibers using ImageJ software.

## ER-to-Golgi and Golgi-to-membrane vesicular trafficking

Low transducibility with adenovirus prevented us from conducting trafficking assays in C2C12 myoblasts or myotubes. Hence, all live cell imaging was performed using primary neonatal rat ventricular myocytes (NRVMs). NRVMs were prepared from 1- to 2-day-old Sprague-Dawley rat pups as previously described (*Lynch et al., 2012*). Exactly 50x10$^3$ cells were plated in either Ibidi μ-slide 8-well dishes (Ibidi USA, Inc. Madison, WI, Cat# 80826) or in 35 mm glass-bottomed laminin-coated MatTek culture dishes (MatTek Corp., P35G-0-10-C) and cultured in HyClone Medium 199/EBSS (ThermoScientific, SH30253FS) supplemented with 2.5% fetal bovine serum (Sigma-Aldrich, F2442) and 1x penicillin-streptomycin (Cellgro 30-0002-CI, Mediatech, Corning Life Sciences, Tewksbury, MA). The next day, NRVMs were infected with adenoviruses harboring the protein of interest (see section adenoviruses, and *Figure 5*) for 3 hr in serum-free media after which they were switched back to culture media supplemented with 2.5% fetal bovine serum.

Life cell quantitative imaging and photobleaching to evaluate ER to Golgi (FRAP) and Golgi to membrane vesicular trafficking (iFRAP) was performed using a Nikon A1 confocal laser microscope system and equipped with Plan Apo 40x oil immersion objective (NA = 1.0), an INU-TIZ-F1 stage top incubator (Tokai hit CO, Ltd, Shizuoka-ken, Japan) and NIS Elements AR microscope imaging software (Nikon Instruments Inc.) as previously described with modifications (*Hirschberg et al., 1998*; *Patterson et al., 2008*; *Zaal et al., 1999*). For ER-to-Golgi protein trafficking experiments, 24 hr after adenoviral infection, Ibidi dishes were infected with CellLight Golgi-RFP Bacmam 2.0 (ThermoFisher Scientific, c10593), a baculovirus containing a fusion construct of human Golgi resident enzyme N-acetylgalactosaminyltransferase and TagRFP (GalNacT2-RFP), according to manufacturer's instructions and incubated overnight. The next day, 100 μg/ml cycloheximide (Sigma-Aldrich, C4859) was added to the NRVMs to block new protein synthesis, 30 min prior to imaging. After acquiring a few baseline images, fluorescence from the Golgi pool of RFP was bleached by irradiating a region of interest (ROI) that encompasses the juxtanuclear Golgi region with a high intensity laser at 561 nm (100% laser power; *Figure 5—figure supplement 1A*). Next, recovery of GalNacT2-

RFP was monitored by time-lapse imaging (5% laser power) at 5-min intervals for 2 hr as a measure of ER to Golgi protein trafficking.

For Golgi to membrane protein trafficking experiments, 24 hr after initial adenoviral infection, NRVMs in MatTek dishes were infected with adenovirus harboring the temperature sensitive VSVG-eGFP and incubated at 40°C for 24 hr to retain the VSVG-eGFP in the ER (*Hirschberg et al., 1998*; *Patterson et al., 2008*). Approximately 60 min prior to imaging, 100 µg/ml cycloheximide was added to the cells. Thirty minutes prior to imaging MatTek dishes were shifted to 32°C allowing the VSVG-eGFP to traffic to the Golgi. After acquiring a few baseline images, the cargo pool in the Golgi was selectively highlighted by photobleaching VSVG-eGFP from the entire cell excluding the perinucelear Golgi network using a high intensity laser at 488 nm (100% laser power; iFRAP; *Figure 5—figure supplement 1B*). Then, time-lapse imaging (5% laser power) at 1-minute intervals for 2 hr was performed to monitor export of VSVG-eGFP molecules from the Golgi using a second area that encompasses the Golgi network as a measurement for Golgi to membrane protein trafficking.

The combination of low energy, high attenuation, and the less concentrated excitation laser beam caused by the low NA objective resulted in negligible photobleaching during repetitive imaging in all experiments. As such, control experiments performing either time-lapsed imaging for 2 hr or the above described FRAP experiment in Golgi-RFP expressing cells in the presence of cycloheximide and brefeldin A (Sigma-Aldrich, B5936; 5 µg/ml) and iFRAP experiment in VSVG-eGFP expressing cells in the presence of cycloheximide and AlF (AlCl$_3$, 60 µM and NaF, 20 µM; 30 min after shift to 32°C) showed no difference in Golgi fluorescence intensity (data not shown) (*Hirschberg et al., 1998*; *Patterson et al., 2008*). Analysis of FRAP and loss of fluorescence after inverse FRAP (iFRAP) experiments was performed as previously described (*Patterson et al., 2008*; *Zaal et al., 1999*). The Golgi fluorescent values were normalized to the average baseline Golgi fluorescence prior to FRAP or to the first data point after iFRAP.

## *Drosophila* genetics and strains

Transgenic UAS-Thbs4 *Drosophila* were generated by P-element-mediated insertion (Rainbow Transgenic Flies, Inc., Camarillo, CA). Briefly, the mouse *Thbs4* cDNA was cloned into the GAL4-responsive pUAST expression vector at the *Eco*RI site to yield the UAS-mThbs4 clone. UAS-mThbs4 transgenic *Drosophila* were generated by microinjection into y[1]w[1118] embryos. The MEF2-Gal4 driver line was obtained from the Bloomington Stock Center (Indiana University, Bloomington, IN). The *Drosophila* model of MD that lacks the δ-sarcoglycan-like gene (Sgcd[840]) and UAS-Tsp *Drosophila* were described previously (*Allikian et al., 2007*; *Subramanian et al., 2007*). The following genotypes were created for the current study: Sgcd[+];UAS-mThbs4/+;MEF2-Gal4/+, Sgcd[840];UAS-mThbs4/+;MEF2-Gal4/+, Sgcd[+];UAS-Tsp/+;MEF2-Gal4/+, Sgcd[840];UAS-Tsp/+;MEF2-Gal4/+, Sgcd[+];UAS-mThbs4/+,Sgcd[+];UAS-Tsp/+ and Sgcd[+];MEF2-Gal4/+. All stocks were raised and maintained on a standard cornmeal-molasses-yeast medium and kept at 25°C on 12:12 light–dark cycle, with 20–40% relative humidity.

## *Drosophila* husbandry and life-span assay

Male *Drosophila* of each genotype were collected at one-day posteclosion. Throughout the study, *Drosophila* were aged at 25°C with a maximum of 12 flies in 25x95 mm polystyrene vials (Fischer Scientific, AS515) and transferred to new vials containing fresh food every three to four days without the use of anesthesia. Survival was recorded every day until 40 days of age. Kaplan–Meier statistical analysis was performed and significance determined by log-rank (Mantel-Cox) tests.

## *Drosophila* negative geotaxis assay

The negative geotaxis assay was performed as previously described (*Allikian et al., 2007*). For each genotyped tested, male *Drosophila* were collected and kept at no more than 12 flies per vial. At 2, 5, 10, 15 and 20 days of age, they were immobilized using CO$_2$, and 12 groups of the various genotypes were transferred into empty polystyrene vials with a line drawn at 80 mm from the base of the vial. (Fisher Scientific, AS515). *Drosophila* were allowed to recover for 1 hr before testing. Each vial was assayed by gently tapping the flies down to the bottom of the vial, thereby engaging their negative geotactic response. The number of *Drosophila* able to climb across the 80-mm against gravity in 10 s was recorded. Four separate trials were performed with a 1-minute resting period in

between. Percentage of *Drosophila* across were averaged and expressed as 'physical performance'. Genotypes were assayed simultaneously to eliminate variability attributed to RT and room humidity.

### *Drosophila* histological analysis and immunohistochemistry

To obtain longitudinal sections of the dorsal median indirect flight muscles, 30-day old *Drosophila* of each genotype were positioned into mounting collars (Genesee Scientific,San Diego, CA 48–100) and subsequently fixed overnight at 4°C in Carnoy's solution (6:3:1 ethanol:chlorophorm:acetic acid), dehydrated, and infiltrated with paraffin. Longitudinal sections were cut at 7 µm thickness and stained for H&E to evaluate dystrophic pathology in the dorsal median indirect flight muscle. To obtain cryosections of the dorsal median indirect flight muscles, the same mounting collars were positioned into O.C.T. freezing medium and placed into liquid nitrogen cooled-isopentane. For Thbs4, Tsp and calreticulin immunohistochemistry, paraffin sections were cleared of paraffin in xylene (2 times for 4 min), rehydrated in ethanol (100% EtOH 2 X 4', 95% EtOH 1 X 3', 70%EtOH 1 X 2', $H_2O$ 1'), incubated for 15 min in PBT (PBS/0.2% Triton X-100) and blocked in PBTB (PBT, 2% BSA) for one hour at RT. The sections were incubated with primary antibody overnight at 4°C (Thbs4: Santa Cruz, Sc-7657-R, 1:50 dilution in PBTB; Tsp, 1/50 [*Subramanian et al., 2007*]; calreticulin: Abcam, ab2907; 1:100 dilution in PBTB), rinsed with PBT and incubated with the appropriate secondary antibody (goat-anti-rabbit-Alexa Fluor-488, Invitrogen; 1:400 in PBTB) for 2 hr at RT along with counterstains. Counterstains included membrane marker WGA conjugated to Alexa Fluor-647 (Invitrogen, W32466; 5 µg/ml, Far Red) and/or a nuclear marker DAPI (Invitrogen, D3571; 1 µg/ml in $H_2O$). For *Drosophila* integrin subunit βPS immunostaining, cryosections were post-fixed in ice-cold methanol for 10 min, washed in PBT, blocked for one hour at room temperature with PBTB and incubated with primary antibody overnight at 4°C (DSHB: CF.6G11; 1:50 in PBTB). Then, tissue was then washed in PBS and subsequently incubated with rabbit-anti-mouse-Alexa Fluor-488 (1:300 in PBTB; Invitrogen) for 2 hr. For each immunostain, consecutive sections were incubated with secondary antibodies alone as a control for specificity (not shown). Images were obtained using a Nikon A1 confocal laser microscope system equipped with 40x $H_2O$ immersion objective (NA = 1.1) and NIS Elements Advanced Research (AR) microscope imaging software (Nikon Instruments Inc.).

### *Drosophila* immunoblotting

For *Drosophila* immunoblotting, twenty 30-day old flies of each genotype were pooled and snap-frozen prior to protein extraction. Total *Drosophila* protein extraction and immunoblotting was essentially performed as described above. Primary antibodies used included: Thbs4 (Santa Cruz, Sc-7657-R; 1:1000); Tsp; 1/100 [*Subramanian et al., 2007*], BiP (GeneTex, GTX48663; 1/1000), calreticulin (Abcam, ab2907; 1:100), PDI (Abcam, ab190883; 1:1000), βPS integrin (DSHB: CF.6G11; 1:100) and β-actin (Abcam, ab8227; 1:2000).

### Transmission electron microscopy

Fresh quadriceps was harvested and immediately immersed in relaxing buffer (0.15% sucrose, 5% dextrose, 100 mM KCl in PBS), subsequently fixed overnight (3.5% glutaraldehyde, 0.15% sucrose in 0.1 M sodium cacodylate ph 7.4) and post-fixed in 1% $OsO_4$ (in water) for 2 hr at RT. Next they were washed, dehydrated and embedded using Epoxy resin. Tissue processing for electron microscopy of the dorsal median inferior flight muscle was performed as previously described (*Allikian et al., 2007*). Briefly, 25 day-old flies were positioned dorsal side up on a spot of OTC freezing medium, dipped in liquid nitrogen, bisected sagittally using a pre-cooled razor and then fixed overnight in 2.5% glutaraldehyde in $NaH_2PO_4$ 0.1 M pH 7.4 at 48°C. Next, they were washed and post-fixed in 2% osmic acid in phosphate buffer for 2 hr at room temperature, dehydrated and embedded using Epon resin. Ultrathin sections of all tissues were counterstained with 1.5% uranyl acetate, 70% ethanol and lead nitrate/Na citrate. Images were obtained using a Hitachi 7600 transmission electron microscope connected to an AMT digital camera.

Sub-sarcolemmal vesicular expansion in myofibers of mouse quadriceps relative to the length of the sarcolemma was determined from 8 to 10 images per myofiber at 2000X of non-overlapping longitudinal regions that were randomly collected. In each image, the size of the vesicular content perpendicular to the sarcolemma -between the sarcolemma and the first sarcomere, and the length of the sarcolemma was determined using ImageJ software. The number of sarcomeric tears per

myofiber in *Drosophila* dorsal median inferior flight muscle was determined by imaging complete myofibers at 1500X from longitudinal sections. Clear disruptions within the sarcomeric structure as indicated in *Figure 8G* and the associated *Figure 8—figure supplement 1G*.

For immunogold labeling of Thbs4, mouse quadriceps and *Drosophila* dorsal median inferior flight muscle were fixed with 4% paraformaldehyde in 0.1 M phosphate buffer (pH 7.4) overnight. After 2 buffer washes, samples were post-fixed with 0.1% $OsO_4$ in the same buffer for 30 min, dehydrated and embedded in hydrophilic acrylic resin following manufacture instruction (L.R. White, #14380; Electron Microscopy Sciences, Hatfield, PA). Ultrathin sections were cut using a Leica Ultra-Cut S or UC6rt ultramicrotome at a thickness of 90 nm and placed on Formvar and carbon coated 200-mesh nickel for immunogold labeling of Thbs4. Briefly, ultrathin sections on grids were first treated with 1% sodium metaperiodate for 60 s to quench $OsO_4$. After several washes with distilled water, grids were placed on drops of PBS containing 5% BSA and 0.1% cold-water fish gelatin to block nonspecific binding. Sections were then incubated overnight at 4°C with goat anti-thrombospondin4 polyclonal primary antibody (R&D Systems, AF2390) at a final concentration of 5 µg/ml. Following several washes, sections were incubated in 6 nm colloidal gold particles conjugated rabbit anti-goat (Electron Microscopy Sciences; #25223) at a concentration of 10–20 µg/ml for 2 hr. After additional washes, all ultrathin sections were stained with 5% uranyl acetate for 2 min and 2% lead citrate for 15 min. For each experiment, both non-transgenic and transgenic sections were labeled and consecutive sections were incubated with secondary antibody alone as a control for specificity (not shown). Immunogold labeling was imaged on a JEOL JEM-1400 transmission electron microscope (JEOL Ltd, Japan) equipped with a Gatan US1000 CCD camera (Gatan, Pleasanton, CA).

## Statistics

All results are presented as mean ± SEM. All data was normally distributed. Statistical analysis was performed with unpaired two-tailed Student's *t* test for two independent groups or one-way ANOVA with *post hoc* Tukey's test for multiple comparisons of 3 or more independent groups, as indicated in the individual figure legends. For survival analysis, Kaplan–Meier statistical analysis was performed and significance was determined by log-rank (Mantel-Cox) tests. All statistics were performed using GraphPad Prism 5.0 for Mac OS X and values were considered statistical significant when $p < 0.05$. No statistical analysis was used to predetermine sample size. The experiments were not randomized and no animals were excluded from analysis. The investigators were not blinded to allocation during experiments and outcome assessment, except for data displayed in *Figure 2B,E*; *Figure 2—figure supplement 1B,C,E,F*; *Figure 2—figure supplement 2D,E*, *Figure 3B,C,F,G*; *Figure 3—figure supplement 1B,C*; *Figure 7F,I*; *Figure 7—figure supplement 2B*; *Figure 8A,B*; *Figure 8—figure supplement 1A,B*. The exact number of animals or biological replicates for each experiment is indicated in the figure legends.

Sample size for the mouse and *Drosophila* experiments were estimated based on previous experiments with similar procedures but also based on past power calculations for appropriate group sizes, and based on this all data reported here were based on adequate sampling. No outlier data were excluded.

## Acknowledgements

This work was supported by grants from the National Institutes of Health (JDM, JER, JD, and EMM), and by the Howard Hughes Medical Institute (JDM). DV, was supported by fellowships from the Research Foundation Flanders, Belgium (FWO-Vlaanderen 1208910N and V4.332.11N) and the Belgian American Educational Foundation, INC. TGS, was supported by a fellowship from the German Research Foundation (Deutsche Forschungsgemeinschaft DFG: SCHI 1290/1-1). MJB, was supported by a training grant from the National Heart, Lung and Blood Institute of the National Institute of Health (F32HL124698). HY was supported in part by the Robert P. Apkarian Integrated Electron Microscopy Core of Emory University and data gathered on the JEOL JEM-1400 120kV TEM by a grant of the National Institute of Health (S10 RR025679).

# Additional information

## Funding

| Funder | Grant reference number | Author |
| --- | --- | --- |
| Research Foundation Flanders | 1208910N | Davy Vanhoutte |
| Research Foundation Flanders | V4.332.11N | Davy Vanhoutte |
| Belgian American Educational Foundation | | Davy Vanhoutte |
| Deutsche Forschungsgemeinschaft | SCHI 1290/1-1 | Tobias G Schips |
| National Institutes of Health | | Jennifer Davis, Joseph E Rabinowitz, Elizabeth M McNally, Jeffery D Molkentin |
| National Heart, Lung, and Blood Institute | F32HL124698 | Matthew J Brody |
| Emory University | | Hong Yi |
| National Institutes of Health | S10 RR025679 | Hong Yi |
| Howard Hughes Medical Institute | | Jeffery D Molkentin |
| National Institutes of Health | P01NS072027 | Jeffery D Molkentin |
| National Institutes of Health | R01HL105924 | Jeffery D Molkentin |

The funders had no role in study design, data collection and interpretation, or the decision to submit the work for publication.

## Author contributions

DV, Conception and design, Acquisition of data, Analysis and interpretation of data, Drafting or revising the article; TGS, JQK, JD, AT, MJB, OK, HY, Acquisition of data, Analysis and interpretation of data; MAS, Performed electron microscopy experiments, Acquisition of data; QQG, EMM, Facilitated experiments by providing Drosophila models, Contributed unpublished essential data or reagents; JER, Acquisition of data, Contributed unpublished essential data or reagents; TV, Analysis and interpretation of data, Contributed unpublished essential data or reagents; JDM, Conception and design, Analysis and interpretation of data, Drafting or revising the article

## Author ORCIDs

Davy Vanhoutte, http://orcid.org/0000-0002-8147-6953
Jeffery D Molkentin, http://orcid.org/0000-0002-3558-6529

## Ethics

Animal experimentation: All animal experiments were approved by the Institutional Animal Care and Use Committee of the Cincinnati Children's Hospital Medical Center (Protocol# IACUC2013-0013). No human subjects or human tissue was directly used in experiments in this study.

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
