## [Decision Letter]

Thank you for submitting your article "Thrombospondin mediates an intracellular vesicular attachment network that stabilizes muscle membranes" for consideration by *eLife*. Your article has been favorably evaluated by Harry Dietz as the Senior Editor and three reviewers, one of whom, Amy Wagers (Reviewer #1), is a member of our Board of Reviewing Editors. The following individual involved in review of your submission has agreed to reveal their identity: Melissa Spencer (Reviewer #3).

The reviewers have discussed the reviews with one another and the Reviewing Editor has drafted this decision to help you prepare a revised submission.

Summary:

This is an interesting study that investigates the role of thrombospondin 4 (Thbs4) in skeletal muscle. The authors report significant alterations in muscle fiber stability and myopathic disease progression in mice lacking Thbs4 and mice with transgenic overexpression of Thbs4 specifically in skeletal muscle. The work parallels to some degree a recent report from the same group on Thbs4 function in cardiac muscle, but the additional analyses of Thbs4 in skeletal muscle, together with the fly analyses, which show congruent phenotypes, are important and provide some novel information.

Overall, the work provides insights into the impacts of thrombospondins in skeletal muscle biology and implicates them as significant contributors to muscle fiber integrity. While all reviewers agreed that the work is appropriate for *eLife*, in principle, several critical concerns emerged upon review that were shared among the reviewers and must be addressed prior to publication (see below).

Essential revisions:

1) The authors appear to have assumed that muscle (or muscle fibers) are the normal site of expression for Thrombospondin; but an alternative interpretation is that the artificial system of overexpression in muscle fibers does not reflect its normal expression pattern. WT muscle fibers do not show any intracellular Thbs4 staining or localization to vesicles suggesting that the intracellular role and protective effects may only occur with Thbs4 overexpression. In Figure 1, staining for Thbs4 in wildtype muscle is not apparent, and it is impossible to tell if the protein is exclusively intracellular or not in Thbs4-Tg or *Scgd^-/-^*, contrary to comments in the text (subsection “Thbs4 augments adaptive ER stress signaling in skeletal muscle and mitigates MD”, second paragraph). Likewise, co-IPs with Thbs4 and β1D-Itg only detect Thbs4/β1D-Itg complexes within vesicles with Thbs4-Tg overexpression, not in WT (Figure 6—figure supplement 1). As such, the manuscript over-interprets results that are based solely on Thbs4 overexpression, and does not acknowledge that these may not apply to normal muscles. The authors should add additional evidence supporting their interpretations, or, if not possible, they should tone down their conclusions and present their discussion in a more balanced manner to highlight this consideration.

2) The manuscript tries to "sell" the paper's novelty by exclusively focusing on intracellular events, and ignores, in comparison, well-established ECM functions. While the intracellular role reported here is interesting and potentially therapeutic, the ECM localization and functions of Thbs have been well established through numerous meticulous studies. This should be acknowledged and the results here presented more accurately as a novel mechanism for ECM proteins inside the cell. The last sentence of the first paragraph in the Discussion, "a growing body of data uniformly suggests that Thbs proteins are not dedicated matricellular or ECM proteins" overstates the case and should be removed. The second paragraph of the Discussion cites a selected set of references without acknowledging many additional papers convincingly showing Thbs4 and other Thbs in ECM (Hauser et al., 1995; Frolova et al. 1995; 2010; 2012; 2014; Christopherson et al., 2005; Sodersten and Ekman, 2006). The authors should revise the manuscript to present a more wholistic and balanced view.

3) Related to the above, co-IPs with Thbs4 and B-integrin show that Thbs4 and β1D-Itg form complexes within vesicles. Are these signaling-ready inside the cell? Are they secreted outside where Thbs4 can localize to the ECM? Is Thbs4 secretion essential for its intracellular functions? Also, if Thbs4 is actively recycled as suggested, then ectopic Thbs4 injected into the ECM should be actively internalized. This should be easily tested.

4) The interpretations of the distinct rescue phenotypes of Thbs4 overexpression versus ATF6α overexpression, together with the vesicle and membrane protein trafficking analyses, seem a bit incongruous. The protective role appears to be specific to Thbs4's interaction with ATF-6. However, ATF-6 levels are not limiting since increasing Thbs4 expression alone can rescue and protect wild-type cells. The early part of the manuscript appears to argue for an important role for ATF6α in Thbs4's functions but data in the latter half of the paper do not support this idea so strongly. Ultimately, the authors conclude that the situation is "complex", and while this is undoubtedly true, and they should be congratulated for including the full complement of their data, even if it cannot be easily explained, the paper would benefit from a more extensive discussion about this and inclusion of possible models that could explain the complexity of the data available thus far. Related to this, a comparison of Thbs4 levels in non-transgenic mice, Thbs4 overexpressors and ATF6α overexpressors could be informative, as the results may reflect a dosage effect for Thbs4.

5) Figure 1. Differences in expression levels between WT and *Sgcd*-null do not seem as great in 1F as in 1B. The authors should quantify the expression data and estimate the degree of overexpression achieved in the transgenic systems (both Thbs4 tg and ATF6α tg). It is also interesting that the induction of ATF6α, BiP, etc. does not seem to scale with Thbs4 expression level, which should be further clarified by quantification of the Western data and discussed in the text.

6) Some of the methods appear incongruous with the way the data is presented. For example, the methods state that AAV-GFP and AAV-Thbs4 were injected into contralateral limbs of the same animal, but this is not mentioned in the text or figure legend. This should be explicitly stated in the figure legend and GFP and Thbs4 levels in injected mice should be compared to uninjected mice to account for possible systemic dissemination of AAVs after intramuscular injection. Also, the authors should clarify if the samples shown are from paired samples from the same mouse, or from different mice.

7) Nuclear position within muscle cells is emphasized throughout the manuscript, but nuclear position is represented only in Figure 3 and there is no discussion of its importance, which should be included.

8) Since transgenic mice were created on FVB/N and *mdx* and *Sgcd*-null are C57 background, the authors should clarify the breeding strategy and number of backcrosses performed to generate the mice used in this study.

9) The paper would benefit greatly from a model figure illustrating the intracellular function of Thbs4 within muscle fibers.

[Editors' note: further revisions were requested prior to acceptance, as described below.]

Thank you for resubmitting your work entitled "Thrombospondin mediates an intracellular vesicular attachment network that stabilizes muscle membranes" for further consideration at *eLife*. Your article has been favorably evaluated by Harry Dietz as the Senior Editor and three reviewers, one of whom, Amy Wagers, is a member of our Board of Reviewing Editors.

We appreciate your responsiveness to prior review. The manuscript clearly has been improved, but there are some remaining issues that need to be addressed before acceptance, as outlined below. We expect that you will be able to address these issues without the need to generate new data.

1) Please add the full analysis of Thbs4 in human muscle disease to the supplement.

2) The possibility that Thbs4 could also be expressed and have a role in non-muscle cells (as discussed in the rebuttal letter) should be mentioned at some point in the manuscript, as this is relevant to interpreting the knockout phenotypes.

3) The title and Abstract should be revised to reflect the fact that, despite addition of new data, the manuscript still cannot unequivocally resolve whether secretion and/or re-uptake essential for Thbs4's "intracellular" functions. Overexpression of Thbs4 with the Thbs4 Tg rescues *mdx* and a *Sgcd* mutants, and this requires Thbs4's interaction with ATF6. ATF6 Tg, as well as a Thbs4 mutant unable to interact with ATF6, both fail to rescue. The authors interpret these and other experiments (e.g. Thbs4-mCa) to mean that the Thbs4 mutant cannot be secreted and is retained in vesicles or in their words that "only the fully membrane-trafficked and secretion competent Thbs4, but not the ER-retained mutant, promoted greater Β1-Itg membrane occupancy" (Response to Reviewers). This would suggest that Thbs4 must be secreted to rescue, potentially supporting an extracellular protective role rather than intracellular. Instead, their interpretation is that it "selectively enhances vesicular trafficking of dystrophin-glycoprotein and Itg attachment complexes to the sarcolemma" or, as stated in the title, that it "mediates an intracellular vesicular attachment network that stabilizes muscle membranes". As the data to support this model do not rule out the alternative, the text describing these results should be more balanced.

4) The authors suggest that there is re-uptake of Thbs4 by myofibers from the myomatrix but it remains unclear if this only occurs upon Thbs4 overexpression. They should add to the Discussion consideration of this issue as an important area for future research – particularly, whether re-uptake of Thbs4 occurs in wild-type myofibers and whether re-uptake may be mediated through Integrins or Dystroglycan.

5) The authors claim that their co-IPs show that Thbs4 in vesicles forms a complex with β1D-Itg receptors in a "signaling-ready complex", but provide no direct evidence. Text should therefore be revised to indicate that signaling competence of this complex remains to be investigated.

6) As stated previously, discussion of Figure 6 should be revised. There appears to be no change in the levels or distribution of B-DG or Dystro in *Sgcd^-/-^* in Figure 6.

7) Authors should comment in the text on why Thbs4 localization appears different between the section co-stained with Calreticulin (new Figure 1, upper right panel) and Collagen I (lower right panel), and why the vesicles appear so different in size and distribution of electron-dense regions between WT and Thbs4 Tg in Figure 1.

---

## [Author Response]

*1) The authors appear to have assumed that muscle (or muscle fibers) are the normal site of expression for Thrombospondin; but an alternative interpretation is that the artificial system of overexpression in muscle fibers does not reflect its normal expression pattern. WT muscle fibers do not show any intracellular Thbs4 staining or localization to vesicles suggesting that the intracellular role and protective effects may only occur with Thbs4 overexpression. In Figure 1, staining for Thbs4 in wildtype muscle is not apparent, and it is impossible to tell if the protein is exclusively intracellular or not in Thbs4-Tg or Scgd^-/-^, contrary to comments in the text (subsection “Thbs4 augments adaptive ER stress signaling in skeletal muscle and mitigates MD”, second paragraph). Likewise, co-IPs with Thbs4 and β1D-Itg only detect Thbs4/β1D-Itg complexes within vesicles with Thbs4-Tg overexpression, not in WT (Figure 6—figure supplement 1). As such, the manuscript over-interprets results that are based solely on Thbs4 overexpression, and does not acknowledge that these may not apply to normal muscles. The authors should add additional evidence supporting their interpretations, or, if not possible, they should tone down their conclusions and present their discussion in a more balanced manner to highlight this consideration.*

We agree with the reviewers that more in-depth analysis were warranted to confirm whether our observations do indeed apply to normal muscle or are solely the result of Thbs4 overexpression. Hence, we performed a series of new experiments to address the concerns raised above.

First, newly included immunofluorescent analysis now reveals that Thbs4 protein is essentially undetectable in uninjured skeletal muscle by this type of assay, while the Thbs4 transgene produced abundant expression that co-localized with calreticulin to a vesicular network on the periphery of the myofibers but was also clearly inside of collagen I staining that marks the extracellular matrix (ECM; Figure 1). However, in diseased skeletal muscle of the *Sgcd^-/-^* mouse, endogenous Thbs4 induction did show protein localization both within the vesicular network inside the myofibers and also within the ECM region in areas of fibrotic accumulation, thus incorporating Thbs4 protein (Figure 1). Similar to our results observed with collagen I, co-labeling with another ECM/matricellular protein periostin again shows that Thbs4 co-localizes to the ECM region in *Sgcd^-/-^* diseased myofibers in areas of fibrosis deposition, but not appreciably in the muscle of Thbs4 TG mice that are uninjured, at least by confocal analysis. (Figure 1 and Figure 1—figure supplement 2). However, by biochemical fractionation of the ECM and subsequent western blotting, which is more sensitive, we can detect some Thbs4 protein in the ECM of these transgenic mice, suggesting that the overexpressed Thbs4 is progressing through the secretory pathway (Figure 1—figure supplement 2).

A description of these newly performed experiments has been added to ‘Histological Analysis and Immunohistochemistry’ subsection of the Materials and methods; these additional findings have been incorporated in the Results section and the data itself is displayed as indicated above in Figure 1 and Figure 1—figure supplement 2.

Second, we investigated whether Thbs4/β1D-Integrin complexes within intracellular vesicles do indeed represent a physiologic process that can be observed in WT and/or *Sgcd^-/-^*muscle. We reasoned that the signal for Thbs4 in the Thbs4 Tg samples of our initial experiments most likely masked any presence of Thbs4 in WT vesicles. Hence, a new set of β1D-integrin positive intracellular vesicles were isolated from WT and *Sgcd^-/-^*quadriceps and then blotted for the presence of Thbs4. This approach revealed that Thbs4, even though lowly expressed in WT mice, does indeed localize to β1D-integrin positive intracellular vesicles in WT and *Sgcd^-/-^*quadriceps and thus represents a physiologic process. See Figure 6—figure supplement 1.

Taken together, these data reveal that our conclusions do indeed apply to normal muscle (patho)physiology and are not the result of the artificial system of overexpression in muscle. This definitely provides an added value to our manuscript; hence we thank the reviewers and editors for their valuable input.

*2) The manuscript tries to "sell" the paper's novelty by exclusively focusing on intracellular events, and ignores, in comparison, well-established ECM functions. While the intracellular role reported here is interesting and potentially therapeutic, the ECM localization and functions of Thbs have been well established through numerous meticulous studies. This should be acknowledged and the results here presented more accurately as a novel mechanism for ECM proteins inside the cell. The last sentence of the first paragraph in the Discussion, "a growing body of data uniformly suggests that Thbs proteins are not dedicated matricellular or ECM proteins" overstates the case and should be removed. The second paragraph of the Discussion cites a selected set of references without acknowledging many additional papers convincingly showing Thbs4 and other Thbs in ECM (Hauser et al., 1995; Frolova et al. 1995; 2010; 2012; 2014; Christopherson et al., 2005; Sodersten and Ekman, 2006). The authors should revise the manuscript to present a more wholistic and balanced view.*

We agree with this comment and have now written the Discussion to be more balanced, with acknowledgement of additional important references that clearly show extracellular functions for Thbs proteins, as suggested by the reviewers. In fact, our newly included confocal data (described in response to essential revision 1) now clearly shows that Thbs4 can accumulate in diseased skeletal muscle from dystrophic mice in areas of fibrotic deposition.

*3) Related to the above, co-IPs with Thbs4 and B-integrin show that Thbs4 and β1D-Itg form complexes within vesicles. Are these signaling-ready inside the cell? Are they secreted outside where Thbs4 can localize to the ECM? Is Thbs4 secretion essential for its intracellular functions? Also, if Thbs4 is actively recycled as suggested, then ectopic Thbs4 injected into the ECM should be actively internalized. This should be easily tested.*

We thank the reviewers for these valuable remarks. As discussed in detail in our response to ‘essential revision 1’ we now provide additional experimental proof that Thbs4 not only localizes to β1D-integrin (Itg) positive intracellular vesicles of our Thbs4 Tg quadriceps, but also in WT and *Sgcd^-/-^*mice (Figure 6—figure supplement 1). Furthermore, Thbs4 is indeed secreted and co-localizes in the collagen I rich extracellular matrix (ECM) surrounding the myofibers of the *Sgcd^-/-^* diseased myofibers, and we can detect a small amount of Thbs4 in a ECM biochemical fractionation preparation, as discussed above (Figure 1 and Figure 1—figure supplement 2). Hence, again our data suggest that these vesicles are indeed signaling ready and traffic through the secretory pathway resulting in incorporation of β1D-integrin in the sarcolemma and secretion of Thbs4 in the ECM (although under nondiseased conditions Thbs4 does not accumulate in the ECM as it is either degraded or taken back up through its recycling receptor; see below).

To test whether Thbs4 secretion essential for its intracellular functions we took advantage of our previously established adenoviral construct encoding a Thbs4 calcium-binding containing mutations in six DXDXDG calcium-binding sites within the T3R domain of mouse Thbs4 (Ad-Thbs4-mCa^2^ ) (Brody et al., 2016). Interestingly, although Thbs4-mCa^2^ contains the wildtype Thbs4 signaling peptide, it is not secreted but rather remains within the ER. However, it does induce an ATF6α mediated ER-stress, both in neonatal rat cardiomyocytes and when expressed in the gastrocnemius muscle of early postnatal rat pups (Brody et al., 2016). Utilizing an identical in vivoapproach with adenoviral gene transfer into the gastrocnemius of early neonatal rat pups, we compared full-length Thbs4 versus the Thbs4-mCa^2^ for effects on β1-Itg membrane levels, as compared to βgal control infection. The data showed that only the fully “membrane-trafficked” and secretion competent Thbs4, but not the ER-retained mutant, promoted greater β1-Itg membrane occupancy (Figure 7—figure supplement 3). Hence, we conclude that Thbs4 must indeed move through the secretory pathway to chaperone at least β1-itg to the sarcolemma.

Next, we addressed the question if Thbs4 is indeed actively recycled; hence potentially explaining the limited amount of Thbs4 deposited in the ECM of our Thbs4 Tg muscles. Unfortunately, we were unsuccessful in reliably injecting recombinant Thbs4 protein only into the ECM of a murine skeletal muscle. Hence we relied on an in vitro approach with standard ‘internalization’ assays to assess uptake of recombinant Thbs4 by cultured C2C12 myoblasts and myotubes as previously described (Chlenski et al., 2011; Nakamura et al., 2014). Interestingly, these in vitroanalysis demonstrated that extracellular Thbs4 is indeed rapidly internalized by both C2C12 myoblasts and myotubes and at least in the case of myoblasts is transported to rab7-positive late endosomes, thus potentially explaining why the presence of Thbs4 protein is low in the ECM of healthy Thbs4 Tg muscles (Figure 1—figure supplement 3). In regards to these findings, we also removed our original statement regarding endocytosis, sorting and recycling of integrins (original submission Figure 6—figure supplement 1), as a more in-depth future analysis would be needed to exclusively focus on Thbs4 and integrin recycling dynamics.

A description of these newly performed experiments discussed above has been added to the Materials and methods section and our novel findings have been incorporated in the Results and Discussion section. The newly generated data itself is displayed as indicated above in Figure 1 and Figure 1—figure supplement 2; Figure 6—figure supplement 1,vE; Figure 7—figure supplement 3 and Figure 1—figure supplement 3.

*4) The interpretations of the distinct rescue phenotypes of Thbs4 overexpression versus ATF6α overexpression, together with the vesicle and membrane protein trafficking analyses, seem a bit incongruous. The protective role appears to be specific to Thbs4's interaction with ATF-6. However, ATF-6 levels are not limiting since increasing Thbs4 expression alone can rescue and protect wild-type cells. The early part of the manuscript appears to argue for an important role for ATF6α in Thbs4's functions but data in the latter half of the paper do not support this idea so strongly. Ultimately, the authors conclude that the situation is "complex", and while this is undoubtedly true, and they should be congratulated for including the full complement of their data, even if it cannot be easily explained, the paper would benefit from a more extensive discussion about this and inclusion of possible models that could explain the complexity of the data available thus far. Related to this, a comparison of Thbs4 levels in non-transgenic mice, Thbs4 overexpressors and ATF6α overexpressors could be informative, as the results may reflect a dosage effect for Thbs4.*

The reviewers raise an important question here. We had not investigated whether the discrepancy of our findings in the ATF6α overexpressors versus the Thbs4 overexpressors could reflect a dosage effect for Thbs4 or induction of ATF6α-dependent ER stress responsive factors. To directly test this, we now performed comparative immunoblot analysis and semi-quantitative analysis of these events (See Figure 7 and Figure 7—figure supplement 1). We now show that skeletal muscle-specific Tg mice overexpressing ATF6α display high levels of nuclear ATF6α protein and induction of BiP, PDI and calreticulin compared to WT mice, all without influencing Thbs4 protein expression levels (Figure 7 and Figure 7—figure supplement 1). Moreover, except for PDI we noted significantly higher levels of these ATF6α-dependent ER stress responsive factors in ATF6α Tg quadriceps as compared to our Thbs4 Tg (Figure 7 and Figure 7—figure supplement 1). Taken together, this indicates that Thbs4-induced ATF6α dependent ER stress itself is not sufficient to produce the membrane stabilizing effect we observed in the Thbs4 Tg mice, even when induced at higher levels than those observed in the Thbs4 Tg itself. However, that Thbs4 itself and its trafficking through the secretory pathway plays a pivotal role in the membrane stabilizing effect of Thbs4 is now better supported in our revised manuscript as discussed above in point #3, where we discuss how Thbs4 must indeed move through the secretory pathway to the membrane to chaperone at least β1 (Figure 7—figure supplement 3).

We agree with the reviewers that adding a model figure that illustrates the intracellular function of Thbs4 within muscle fibers would be a great added value to our manuscript. Hence, newly included Figure 9 now provides a comprehensive model of our proposed intracellular mechanism, in addition to models that illustrate the differences we observe between when overexpressing Thbs4 as compared to ATF6α overexpressing muscle.

*5) Figure 1. Differences in expression levels between WT and Sgcd-null do not seem as great in 1F as in 1B. The authors should quantify the expression data and estimate the degree of overexpression achieved in the transgenic systems (both Thbs4 tg and ATF6α tg). It is also interesting that the induction of ATF6α, BiP, etc. does not seem to scale with Thbs4 expression level, which should be further clarified by quantification of the Western data and discussed in the text.*

In agreement with the reviewers remark we have now quantified the expression levels of the immunoblots Figure 1 and present these present this analysis in a newly added Figure 1—figure supplement 1.

Furthermore, as extensively discussed in response to the ‘essential revision 4’, Figure 7 and Figure 7—figure supplement 1 now includes new immunoblots and their semi-quantitative analysis that estimates the degree of Thbs4 and nuclear ATF6α achieved in both Thbs4 Tg and ATF6α Tg quadriceps and their induction of Bip, PDI and calreticulin. These new results have been discussed in the Results section. As indicated above, the newly generated data itself can be found in Figure 1—figure supplement 1, Figure 7 and Figure 7—figure supplement 1.

*6) Some of the methods appear incongruous with the way the data is presented. For example, the methods state that AAV-GFP and AAV-Thbs4 were injected into contralateral limbs of the same animal, but this is not mentioned in the text or figure legend. This should be explicitly stated in the figure legend and GFP and Thbs4 levels in injected mice should be compared to uninjected mice to account for possible systemic dissemination of AAVs after intramuscular injection. Also, the authors should clarify if the samples shown are from paired samples from the same mouse, or from different mice.*

We would like to apologize to the reviewers if description of the AAV9 experimental approach in the original manuscript was unclear. We agree with the reviewer’s comment that if AAV9-eGFP and AAV9-Thbs4 were to be injected into contralateral limbs from the same animal that there could (i) be potential systemic dissemination of AAVs after intramuscular injection and that (ii) additional experimental controls such as untreated muscles would be appropriate. However, our original submitted data was derived from mice in which we did not mix the AAVs in a single animal. In other words, we treated both left and right gastrocnemius muscles of 3-day old *Sgcd^-/-^*mice with either AAV9-eGFP or in separate pups, only AAV9-Thbs4. Mice were sacrificed at 6 weeks of age. The left gastrocnemius of each mouse was fixed, processed, paraffin embedded, sectioned and stained with H&E and Masson’s Trichrome, whereas the right muscles were snap-frozen in liquid nitrogen for storage at -80°C. Hence the samples shown in the figure are from different mice.

To avoid potential confusion for the reader regarding our experimental setup, we now specifically clarify our approach in the Results section, the ‘Adeno-associated virus (AAV) Serotype-9 Production and injection’ subsection of the Materials and methods and the figure legend of Figure 2—figure supplement 4.

*7) Nuclear position within muscle cells is emphasized throughout the manuscript, but nuclear position is represented only in Figure 3 and there is no discussion of its importance, which should be included.*

In agreement with the reviewer’s comment, the revised manuscript now includes (i) a new Figure 2—figure supplement 3 displaying H&E histological assessment of central nucleation from quadriceps with Thbs4 overexpression or when crossed into both Sgcd^-/-^ and mdx dystrophic backgrounds; (ii) H&E images were added to Figure 2—figure supplement 4, and Figure 7—figure supplement 2; and (iii) discussion of what central nucleation represents is also now given in the revised paper.

*8) Since transgenic mice were created on FVB/N and mdx and Sgcd-null are C57 background, the authors should clarify the breeding strategy and number of backcrosses performed to generate the mice used in this study.*

In agreement with the reviewer’s comment, we expanded the ‘mouse models’ subsection of the Materials and methods with the breeding strategy and number of backcrosses performed to generate the mice used in our study.

More specifically, the ‘Mouse Models’ subsection now state: “Next, Ska-Thbs4-Tg, ska-ATF6α-Tg and Thbs4^-/-^ mice were backcrossed for at least 6 generations into the *Sgcd^-/-^* background to generate *Sgcd^-/-^* Thbs4-Tg mice, *Sgcd^-/-^* ATF6α-Tg mice and *Sgcd^-/-^ Thbs4^-/-^* mice, as well as their littermate controls. An identical breeding strategy was used to generate mdx *Thbs4^-/-^* mice. In addition, males from each transgenic line were crossed to mdx heterozygous females to generate *mdx*-Tg and *mdx* non-Tg male littermates and their appropriate controls.”

*9) The paper would benefit greatly from a model figure illustrating the intracellular function of Thbs4 within muscle fibers.*

As indicated in the response to ‘essential revision 4’, we agree with the reviewers that adding a model figure that illustrates the intracellular function of Thbs4 within muscle fibers would be a great added value to our manuscript. Hence, newly included Figure 9 now provides a comprehensive model of our proposed intracellular mechanism. In addition, this figure and accompanying figure legend also includes models to illustrate the differences we observe between when overexpressing Thbs4 as compared to ATF6α overexpressing muscle.

[Editors' note: further revisions were requested prior to acceptance, as described below.]

*1) Please add the full analysis of Thbs4 in human muscle disease to the supplement.*

In agreement with the reviewers remark we now display the analysis of Thbs4 mRNA relative expression levels in all 11 human muscle diseases in Figure 1—figure supplement 1. In addition, the ‘Thbs4 augments adaptive ER stress signaling in skeletal muscle and mitigates MD’subsection of the Results section and figure legends for Figure 1 and Figure 1—figure supplement 1 have been adjusted accordingly.

*2) The possibility that Thbs4 could also be expressed and have a role in non-muscle cells (as discussed in the rebuttal letter) should be mentioned at some point in the manuscript, as this is relevant to interpreting the knockout phenotypes.*

As correctly pointed out by the reviewers our current findings do not rule out other, non-muscle cells as other potential cellular sources of Thbs4 expression and secretion in skeletal muscle. As requested, following statement was added to the *‘*Thbs4 augments adaptive ER stress signaling in skeletal muscle and mitigates MD’subsection of the Results section: “Hence, although our findings currently do not rule out that other, non-muscle cells in thedystrophic muscle might express and secrete Thbs4, our data collectively identify the myofiber as an important cellular source of Thbs4 expression, secretion and re-uptake.”

*3) The title and Abstract should be revised to reflect the fact that, despite addition of new data, the manuscript still cannot unequivocally resolve whether secretion and/or re-uptake essential for Thbs4's "intracellular" functions. Overexpression of Thbs4 with the Thbs4 Tg rescues mdx and a-Sgcd mutants, and this requires Thbs4's interaction with ATF6. ATF6 Tg, as well as a Thbs4 mutant unable to interact with ATF6, both fail to rescue. The authors interpret these and other experiments (e.g. Thbs4-mCa) to mean that the Thbs4 mutant cannot be secreted and is retained in vesicles or in their words that "only the fully membrane-trafficked and secretion competent Thbs4, but not the ER-retained mutant, promoted greater Β1-Itg membrane occupancy" (Response to Reviewers). This would suggest that Thbs4 must be secreted to rescue, potentially supporting an extracellular protective role rather than intracellular. Instead, their interpretation is that it "selectively enhances vesicular trafficking of dystrophin-glycoprotein and Itg attachment complexes to the sarcolemma" or, as stated in the title, that it "mediates an intracellular vesicular attachment network that stabilizes muscle membranes". As the data to support this model do not rule out the alternative, the text describing these results should be more balanced.*

Although we believe that our data make a strong case that Thbs4 does indeed mediate an intracellular vesicular attachment network that stabilizes muscle membranes from inside the myofiber, there are rarely absolutes and our findings do not definitively demonstrate an exclusive intracellular function for Thbs4. Hence, as suggested we have changed the title and Abstract of the revised paper to remove the implication that Thbs4 functions intracellularly. Further studies are currently ongoing and will hopefully shed an unequivocal light on this pressing question. The exact changes are:

Title: “Thrombospondin mediates an intracellular vesicular attachment network that stabilizes muscle membranes” to “Thrombospondin expression in myofibers stabilizes muscle membranes”.

Revised the last sentence of the Abstract from “This functional conservation emphasizes the fundamental importance of Thbs’ asintracellular regulators of cellular attachment and membrane stability and identifies Thbs4 as a potential therapeutic target for muscular dystrophy” to “This functional conservation emphasizes the fundamental importance of Thbs’ as regulators of cellular attachment and membrane stability and identifies Thbs4 as a potential therapeutic target for muscular dystrophy.”

*4) The authors suggest that there is re-uptake of Thbs4 by myofibers from the myomatrix but it remains unclear if this only occurs upon Thbs4 overexpression. They should add to the Discussion consideration of this issue as an important area for future research – particularly, whether re-uptake of Thbs4 occurs in wild-type myofibers and whether re-uptake may be mediated through Integrins or Dystroglycan.*

We thank the reviewers for their constructive comment. We agree that this finding opens up an important area for future research and have therefore extended the Discussion of our manuscript on the re-uptake of Thbs4 into the myofiber with the following consideration:

“These dynamic localization differences at baseline versus during fibrotic disease could be attributed to Thbs recycling at the cell surface (Adams and Lawler, 2011; Wang et al., 2004). Indeed, we observed rapid up take of recombinant Thbs4 when given exogenously to cultured C2C12 myoblasts or myotubes. It is possible that extracellular Thbs4 reuptake could occur through its designated receptor or through the integrin and DGC complexes. Indeed, other matricellular proteins such as SPARC were shown to be actively taken up back into myofibers through such a process with integrin associated endocytosis, sorting and recycling (Chlenski et al., 2011; De Franceschi et al., 2015; Nakamura et al., 2014).”

*5) The authors claim that their co-IPs show that Thbs4 in vesicles forms a complex with β1D-Itg receptors in a "signaling-ready complex", but provide no direct evidence. Text should therefore be revised to indicate that signaling competence of this complex remains to be investigated.*

As pointed out by the reviewers our data indeed do not provide direct evidence to support that Thbs4 in vesicles forms a complex with β1D-integrin in a ‘signaling-ready complex’. Hence, we have revised our original statement in the Discussion of the manuscript:

“Our ultrastructural and biochemical analyses showed that the Thbs-integrin complex resides withinthe lumen of vesicles, likely in their preassembly stage on their way to the cell surface.”

to:

“Our ultrastructural and biochemical analyses showed that the Thbs-integrin complex resides withinthe lumen of vesicles. Although further, in-depth analyses will suggest whether Thbs4 co-regulates the signaling competence of this integrin complex and the exact preassembly stage that is influenced by Thbs4 on the way to the cell surface.”

*6) As stated previously, discussion of Figure 6 should be revised. There appears to be no change in the levels or distribution of B-DG or Dystro in Sgcd^-/-^ in Figure 6.*

We apologise to the reviewers for how we reworded this sentence and we agree it was not very clear. In agreement with the reviewers comment we have now corrected our original discussion of the results presented in Figure 6, stating “As previously observed, loss of δ-sarcoglycan in skeletal muscle of *Sgcd^-/-^* mice resulted in the near complete absence of the other sarcoglycans at the membrane (as they all form a complex), with reductions in dystroglycan and dystrophin (Figure 6) (Durbeej and Campbell, 2002).”

to:

“As previously observed, loss of δ-sarcoglycan in skeletal muscle of *Sgcd^-/-^* mice resultedin the near complete absence of the other sarcoglycans at the membrane (as they all form a complex; Figure 6) (Durbeej and Campbell, 2002).”

*7) Authors should comment in the text on why Thbs4 localization appears different between the section co-stained with Calreticulin (new Figure 1, upper right panel) and Collagen I (lower right panel), and why the vesicles appear so different in size and distribution of electron-dense regions between WT and Thbs4 Tg in Figure 1.*

We thank the reviewers for this valuable comment. Unfortunately, we were unable to co- label tissue sections with Thbs4 and ER marker calreticulin, on the one hand, or Thbs4 and collagen I or periostin, on the other hand, using a singular tissue processing method. As originally described in detail in the Materials and methods section co-staining with calreticulin occurred on paraffin-embedded muscle sections, whereas co-staining with either collagen I or periostin was performed using cryo-embedded tissue sections. Hence, we believe that Thbs4 localization appears slightly different due to differences in tissue processing and subsequent Thbs4 epitope availability for antibody binding. To clarify and further underscore this for the reader, we have revised our original Results section:

“Tissue Immunofluorescent analysis revealed that Thbs4 protein was undetectableinuninjured skeletal muscle while the transgene produced abundant expression that co-localized with calreticulin to a vesicular network on the periphery of the myofibers that was also clearly inside of collagen I staining that marks the ECM (Figure 1). However, in diseased skeletal muscle of the *Sgcd^-/-^* mouse, endogenous Thbs4 induction showed protein localization again within the vesicular network inside the myofibers, but also within regions of ECM that had fibrotic tissue deposition (Figure 1).”

to:

"Tissue Immunofluorescent analysis revealed that Thbs4 protein was undetectable inuninjured skeletal muscle while the transgene produced abundant expression that co-localized with calreticulin to a vesicular network on the periphery of the myofibers of paraffin-embedded quadriceps and was also clearly inside of collagen I staining that marks the ECM of cryo- embedded quadriceps (Figure 1). Furthermore, although Thbs4 protein localization appeared slightly different between paraffin- and cryo-embedded skeletal muscle of the *Sgcd^-/-^* mouse, induction of endogenous Thbs4 again showed localization within the vesicular network inside the myofibers, and only within limited regions outside of myofibers where fibrotic tissue deposition was prominent (Figure 1)."

Secondly, using transmission electron microscopy and immunogold detection we do indeed consistently observe that overexpression of Thbs4 induces a very large expansion of seemingly homogenously smaller sized, Thbs4 dens, sub-sarcolemmal and intramyofibrillar ER and post-ER vesicles, as compared to the irregularly sized, less electron-dense, Thbs4 containing vesicles observed in WT muscle (Figure 1; Figure 1—figure supplement 2). Although highly interesting, the molecular machinery that determines vesicular size, in general, and in our case is driving this altered vesicular phonotype observed upon Thbs4 overexpression is poorly understood and currently still under investigation. Nevertheless, to highlight these findings and our future efforts, we have revised the original discussion of these results from:

“Remarkably, Thbs4 overexpression in skeletal muscle caused a dramatic induction *of* sub-sarcolemmal and intramyofibrillar ER and post-ER vesicles that also contain Thbs4, as shown by transmission electron microscopy and immunogold detection (Figure 1; Figure 1—figure supplement 2). Thus, Thbs4 overexpression in skeletal muscle produces a dramatic expansionof the intracellular vesicular network and an ER stress response.”

to:

“Remarkably, transmission electron microscopy and immunogold detection revealed thatThbs4 overexpression in skeletal muscle caused a dramatic induction of sub-sarcolemmal and intramyofibrillar ER and post-ER vesicles that contained Thbs4 protein (Figure 1; Figure 1—figure supplement 2). These Thbs4-dependent vesicles were highly uniform in size and moreelectron dense compared with similar vesicles in subsarcolemmal regions from WT muscle. Future studies will investigate the nature of these Thbs4-expanded vesicles and their composition based on known variables (Malhotra and Erlmann, 2015; Paczkowski et al., 2015)."